# Identification of purine biosynthesis as an NADH-sensing pathway to mediate energy stress

Ronghui Yang[1,2,3], Chuanzhen Yang[1,2,3], Lingdi Ma[1], Yiliang Zhao[1], Zihao Guo[1], Jing Niu[1], Qiaoyun Chu[1], Yingmin Ma[2] & Binghui Li [1,2] ✉

An enhanced NADH/NAD+ ratio, termed reductive stress, is associated with many diseases. However, whether a downstream sensing pathway exists to mediate pathogenic outcomes remains unclear. Here, we generate a soluble pyridine nucleotide transhydrogenase from *Escherichia coli* (*Ec*STH), which can elevate the NADH/NAD+ ratio and meantime reduce the NADPH/NADP+ ratio. Additionally, we fuse *Ec*STH with previously described *Lb*NOX (a water-forming NADH oxidase from *Lactobacillus brevis*) to resume the NADH/NAD+ ratio. With these tools and by using genome-wide CRISPR/Cas9 library screens and metabolic profiling in mammalian cells, we find that accumulated NADH deregulates PRPS2 (Ribose-phosphate pyrophosphokinase 2)-mediated downstream purine biosynthesis to provoke massive energy consumption, and therefore, the induction of energy stress. Blocking purine biosynthesis prevents NADH accumulation-associated cell death in vitro and tissue injury in vivo. These results underscore the pathophysiological role of deregulated purine biosynthesis in NADH accumulation-associated disorders and demonstrate the utility of *Ec*STH in manipulating NADH/NAD+ and NADPH/NADP+.

Reductive stress, reflected in the elevated intracellular NADH/NAD+ ratio or NADH accumulation, is associated with multiple human diseases, such as cancer, neurodegeneration, diabetes, and mitochondrial disorders[1–5]. These syndromes often feature a hypoxic micro-environment or ETC impairment. Although such impediments to respiration usually lead to many biological or physiological outcomes, recent reports have demonstrated that NADH accumulation exerts a critical role in determining cell growth arrest or cell death[5–8]. NADH accumulation can extensively drive metabolic reprogramming to dissipate electrons in cells under stressful conditions, which usually hinges on anabolic processes, in particular lipogenesis that assimilates electrons in the form of NADPH[6,9–11]. Such an adaptive mechanism widely exists in proliferating cancer cells under hypoxia or ETC inhibition[6]. Therefore, NADH-reductive stress could include a decreased NADPH level and NADPH/NADP+ ratio.

However, as for non-proliferating differentiated cells, an increase in the NADH/NAD+ ratio resulting from hypoxia or pathological impairment of ETC could potentially cause health problems, such as ischemic injury, neurodegenerative diseases, and mitochondrial myopathies[12,13]. Therefore, to ameliorate reductive stress, the development of strategies designated to reduce the NADH/NAD+ ratio is now emerging as an important disease-modifying process. Pyruvate can be converted to lactate using lactate dehydrogenase, and this process consumes NADH. Although it reduces cellular NADH substantively in vitro, it is ineffective in vivo[4]. Recently, an engineered fusion (LOXCAT) between bacterial lactate oxidase and catalase, which irreversibly converts lactate and oxygen to pyruvate and water, has been found to reduce the NADH/NAD+ ratio in vitro and in vivo[4]. In addition, a water-forming NADH oxidase from *Lactobacillus brevis* (*Lb*NOX) also can effectively prevent NADH accumulation in vitro and in vivo[3,5]. However, how accumulated NADH triggers pathogenic

[1]Department of Biochemistry and Molecular Biology, Capital Medical University, Beijing, China. [2]Beijing Institute of Hepatology, Beijing Youan Hospital, Capital Medical University, Beijing, China. [3]These authors contributed equally: Ronghui Yang, Chuanzhen Yang. ✉e-mail: bli@ccmu.edu.cn

outcomes remains to be elucidated. Hence, if we can determine effector pathways associated with reductive stress or NADH accumulation, we may provide more targets for the treatment of these types of metabolic disorders.

In the current study, we develop a genetic tool to manipulate the ratios of cellular NADH/NAD$^+$ and NADPH/NADP$^+$. With the assistance of this tool, we identify PRPS2-mediated purine biosynthesis as an accumulated NADH-sensing pathway to induce energy stress using genome-wide CRISPR/Cas9 library screens in combination with metabolic profiling. Moreover, we demonstrate that blocking purine biosynthesis significantly prevents tissue injury induced by reductive stress resulting from increased metabolic oxidation in a mouse model.

## Results

### Generation of tools for manipulation of cellular NADH/NAD$^+$ and NADPH/NADP$^+$

Electrons in NADH and NADPH can be translocated or trans hydrogenated between each other[6,14]. Therefore, the couples of NADH/NAD$^+$ and NADPH/NADP$^+$ are tightly metabolically connected in cells (Fig. 1a). Upon reductive stress, the elevated NADH/NAD$^+$ ratio can rewire cellular transformations to dissipate electrons mainly by promoting glutamine-initiated lipogenesis, thus it often leads to a concomitant decrease in the NADPH/NADP$^+$ ratio[6]. Typically, hypoxia and ETC inhibition by antimycin A enhanced the cellular level of NADH and ratio of NADH/NAD$^+$, concomitantly reducing the cellular level of NADPH and ratio of NADPH/NADP$^+$ (Supplementary Fig. 1a, b). If we used α-ketobutyrate, a pyruvate analog that can be reduced to excretory α-hydroxybutyrate and thus reduce NADH accumulation[8], to treat cells, it simultaneously restored the ratios of NADH/NAD$^+$ and NADPH/NADP$^+$ and cell proliferation under the conditions of hypoxia and ETC inhibition (Supplementary Fig. 1a–c).

To mimic NADH-accumulation associated with NADPH decrease, we tried to use a soluble pyridine nucleotide transhydrogenase (STH) from *Escherichia coli* (*Ec*STH) or *Pseudomonas fluorescens* (*Pf*STH) as the genetic tool, that can transfer electrons from NADPH to NAD$^+$ and produce NADP$^+$ and NADH[15–17] (Fig. 1a). We generated a HeLa cell line to express an N-terminal or C-terminal Flag-tagged *Ec*STH or *Pf*STH under the control of a doxycycline (Dox)-inducible Tet-on promoter (TRE3G) (Fig. 1b and Supplementary Fig. 2a). Expression of *Ec*STH or *Pf*STH led to an increase in the ratio of NADH/NAD$^+$, with a concomitant decrease in the ratio of NADPH/NADP$^+$, while EGFP expression did not at all (Fig. 1c and Supplementary Fig. 2b–d). We further fused *Ec*STH or *Pf*STH to *Lb*NOX with three tandem flagged peptides as the linker (Fig. 1d, e; Supplementary Fig. 2e). *Lb*NOX can oxidize NADH to NAD$^+$ using oxygen as the electron receptor (Fig. 1a), and it does not affect cell proliferation but prevents NADH accumulation[5]. As expected, *Lb*NOX fusion restored the cellular NADH level and NADH/NAD$^+$ ratio, but not the NADHP level and NADPH/NADP$^+$ ratio, induced by *Ec*STH or *Pf*STH (Fig. 1f and Supplementary Fig. 2f–h). Among the fusions, *Ec*STH-*Lb*NOX worked most effectively and restored the NADH/NAD$^+$ ratio to the best extent (Fig. 1f). Therefore, *Ec*STH-Flag (*Ec*STH) and *Ec*STH-*Lb*NOX were used in the ongoing experiments, and they exhibited diffuse localization in the cytoplasm (Supplementary Fig. 2i, j).

Our results further showed that *Ec*STH, not *Ec*STH-*Lb*NOX or EGFP, significantly increased the ratios of secreted lactate/pyruvate and β-hydroxybutyrate/acetoacetate (Fig. 1g), traditionally used as proxies for the cytosolic and mitochondrial NADH/NAD$^+$ ratios[18], respectively, confirming again the induction of reductive stress. We then purified *Ec*STH, *Lb*NOX and *Ec*STH-*Lb*NOX proteins (Fig. 2a), and measured their kinetic parameters. Our results showed that $V_{max}$, $K_M$ and $k_{cat}$ values of *Ec*STH were comparable to those of *Ec*STH in *Ec*STH-*Lb*NOX (Fig. 2b–e). However, $V_{max}$ and $k_{cat}$ of *Lb*NOX were much

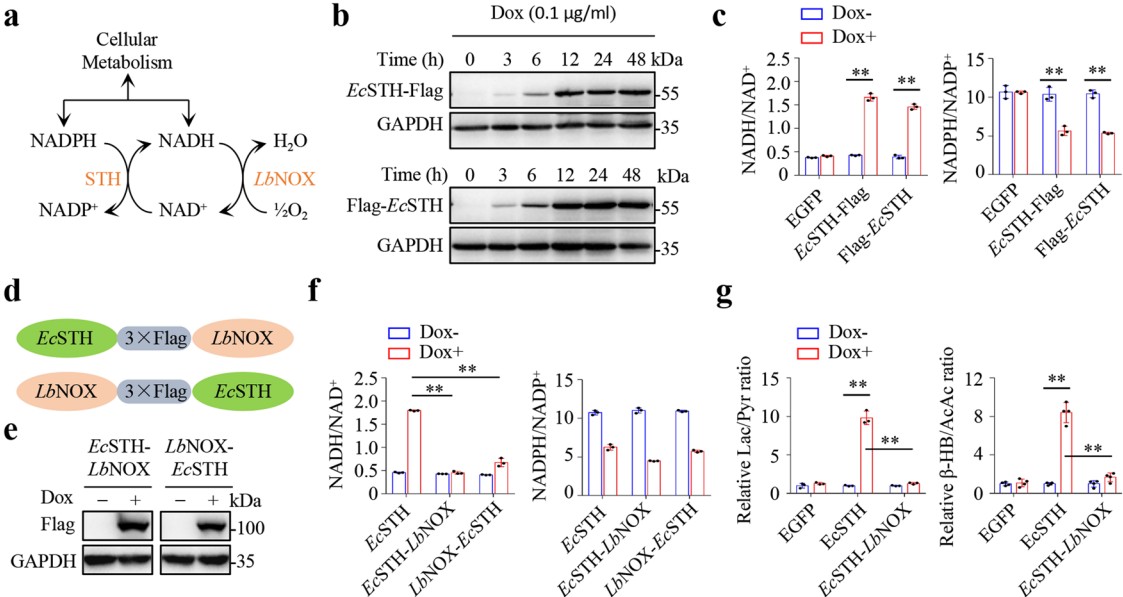

**Fig. 1 | Generation of tools to manipulate cellular NADH/NAD+ and NADPH/NADP+. a** Enzymatic reactions catalyzed by STH and *Lb*NOX. **b** Western blots for the expression of inducible Tet-on Flag-tagged *Ec*STH in HeLa cells induced by Dox (0.1 μg/mL) for different times. **c** The ratios of cellular NADH/NAD$^+$ and NADPH/NADP$^+$ in HeLa cells expressing Tet-on Flag-tagged *Ec*STH cultured with Dox (0.1 μg/mL) for 12 h (*n* = 3 biologically independent samples). From left to right: **P* = 1.08e-05, **P* = 1.10e-05, **P* = 0.0015, **P* = 6.88e-05. **d** Schematic for the fusion of *Ec*STH and *Lb*NOX. **e** Western blots for the expression of inducible Tet-on fusions of *Ec*STH with *Lb*NOX in HeLa cells induced by Dox (0.1 μg/mL) for 12 h. **f** The ratios of cellular NADH/NAD$^+$ and NADPH/NADP$^+$ in HeLa cells

expressing Tet-on fusion of *Ec*STH with *Lb*NOX cultured with Dox (0.1 μg/mL) for 12 h (*n* = 3 biologically independent samples). From left to right: **P* = 2.03e-07, **P* = 2.64e-05. **g** Secretory lactate/pyruvate (Lac/Pyr, *n* = 3 biologically independent samples) and β-hydroxybutyrate/acetoacetate (β-HB/AcAc, *n* = 4 biologically independent samples) of HeLa cells expressing inducible Tet-on EGFP, *Ec*STH or *Ec*STH-*Lb*NOX cultured with Dox (0.1 μg/mL) for 12 h. From left to right: **P* = 6.50e-05, **P* = 7.62e-05, **P* = 8.37e-06, **P* = 2.29e-05. All experimental data were verified in at least three independent experiments. Error bars represent mean ± SD. **P* < 0.01; Two-tailed Student's *t*-tests. Source data are provided as a Source Data file.

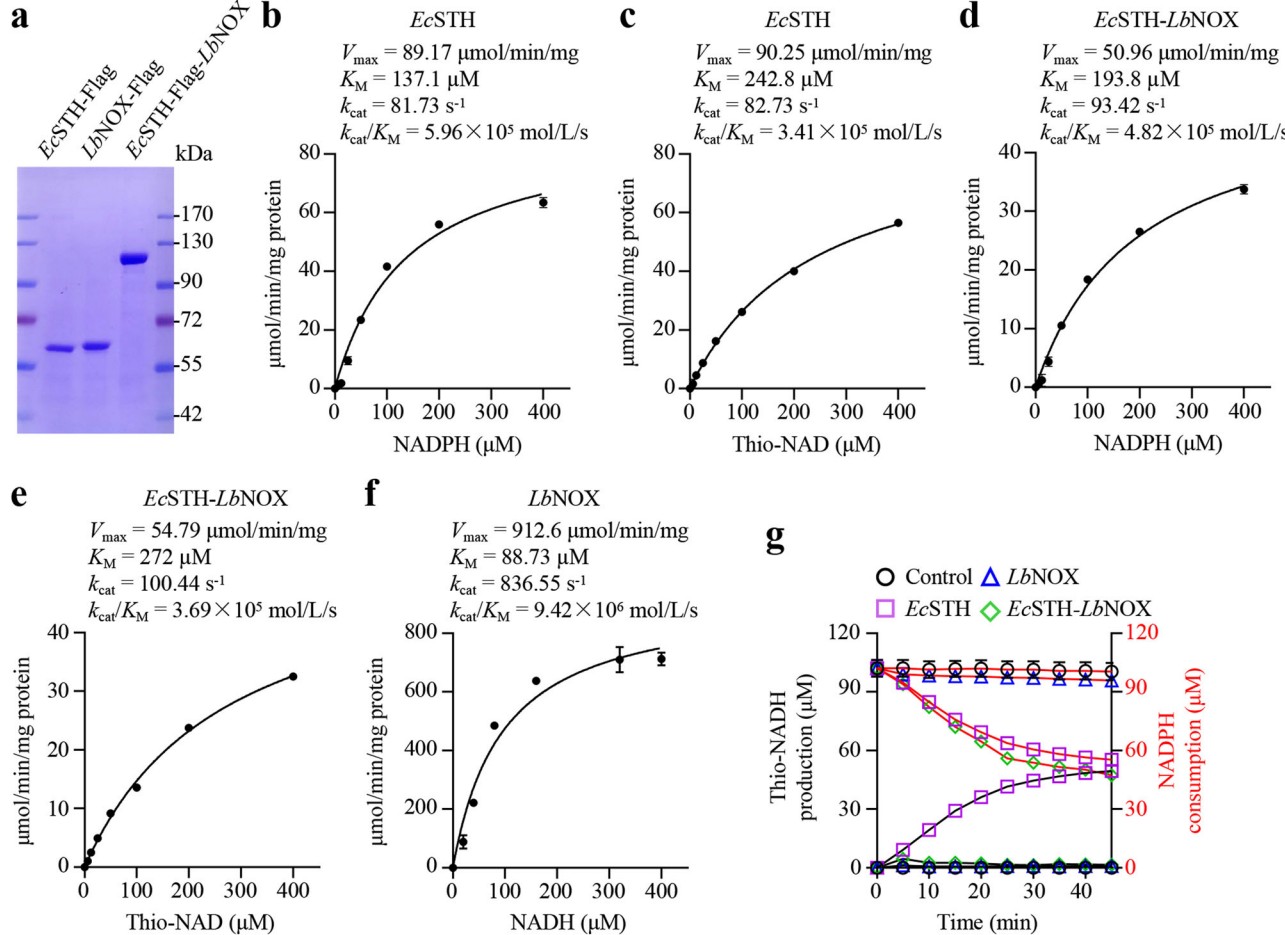

**Fig. 2 | The enzyme kinetics of LbNOX, EcSTH, and EcSTH-LbNOX. a** SDS-PAGE of purified recombinant *Ec*STH, *Lb*NOX, and *Ec*STH-*Lb*NOX. **b** The activity of *Lb*NOX in relation to NADH concentration ($n = 3$ biologically independent samples). **c** The activity of *Ec*STH in relation to NADPH concentration (with 400 μM thio-NAD⁺) ($n = 3$ biologically independent samples). **d** The activity of *Ec*STH in relation to Thio-NAD⁺ concentration (with 400 μM NADPH) ($n = 3$ biologically independent samples). **e** The activity of *Ec*STH-*Lb*NOX in relation to NADPH concentration (with 400 μM thio-NAD⁺) ($n = 3$ biologically independent samples). **f** The activity of

*Ec*STH-*Lb*NOX in relation to Thio-NAD⁺ concentration (with 400 μM NADPH) ($n = 3$ biologically independent samples). **g** Simultaneous measurement of the reduction of thio-NAD⁺ and the oxidation of NADPH by *Ec*STH, *Lb*NOX, and *Ec*STH-*Lb*NOX (with 400 μM thio-NAD⁺ and NADPH respectively) ($n = 3$ biologically independent samples). All experimental data were verified in at least two independent experiments. Error bars represent mean ± SD. Source data are provided as a Source Data file.

greater than those of *Ec*STH (Fig. 2f), so that *Ec*STH-*Lb*NOX was unable to produce NADH while only consuming NADPH (Fig. 2g), consistent with the results obtained from *Ec*STH-*Lb*NOX expressed in cells (Fig. 1f and Supplementary Fig. 2f).

## *Ec*STH causes growth inhibition and cell death by inducing NADH accumulation

Furthermore, we observed that the extent to which *Ec*STH altered the level or ratio of NAD(P)H/NAD(P)⁺ increased with Dox treatment in a dose-dependent manner (Fig. 3a, b; Supplementary Fig. 3a, b). *Ec*STH mainly increased the content of NADH but affected the cellular level of NAD⁺ to a lesser extent (Supplementary Fig. 3a).

*Ec*STH expression significantly suppressed cell proliferation and colony formation, and even killed cells upon induction with 1 μg/mL Dox (Fig. 3c, d and Supplementary Fig. 3c). By contrast, the expression of EGFP or *Lb*NOX did not affect cell growth at any concentration of Dox (Fig. 3c, d). Moreover, the inhibitory effects on cell growth afforded by *Ec*STH were completely reversed by *Lb*NOX fusion (Fig. 3c, d). Correspondingly, *Ec*STH also significantly suppressed tumor growth in a xenograft model while *Ec*STH-*Lb*NOX did not (Fig. 3e). Furthermore, our results showed that α-ketobutyrate indeed reduced the ratio of NADH/NAD⁺ (Supplementary Fig. 3d),

and dramatically prevented death in cells expressing *Ec*STH even when induced by 1 μg/mL Dox (Fig. 3f). In addition, *Ec*STH-induced cell death was significantly prevented by z-VAD-FMK (a pan-inhibitor of caspases), but not by necrosulfonamide (NSA, an inhibitor of necroptosis) and ferrostatin-1 (an inhibitor of ferroptosis) (Fig. 3g), suggesting the occurrence of apoptosis. It was further confirmed by the results that *Ec*STH expression promoted Annexin V-positive cells (Fig. 3h and Supplementary Fig. 3e), and enhanced the levels of cleaved caspase-3 and PARP1, which was prevented by the addition of α-ketobutyrate (Fig. 3i).

Now, to test the generality of *Ec*STH and *Ec*STH-*Lb*NOX, we inducibly expressed them in another cancer cell line, MDA-MB-231, and mouse embryonic fibroblasts (MEF) (Supplementary Figs. 4a and 5a), and obtained the similar results. *Ec*STH expression increased the level of NADH and ratio of NADH/NAD⁺, while decreasing the ratio of NADPH/NADP⁺ (Supplementary Figs. 4b, 4c, 5b, and 5c). *Lb*NOX fusion restored the homeostasis of NADH/NAD⁺, but not that of NADPH/NADP⁺ (Supplementary Figs. 4b, 4c, 5b, and 5c). Similarly, *Ec*STH, not *Ec*STH-*Lb*NOX, suppressed cell proliferation or elicited cell death in MDA-MB-231 and MEF cells (Supplementary Figs. 4d and 5d). The electron acceptor, α-ketobutyrate, also rescued proliferation (Supplementary Figs. 4e and 5e) and prevented apoptosis of MDA-MB-231

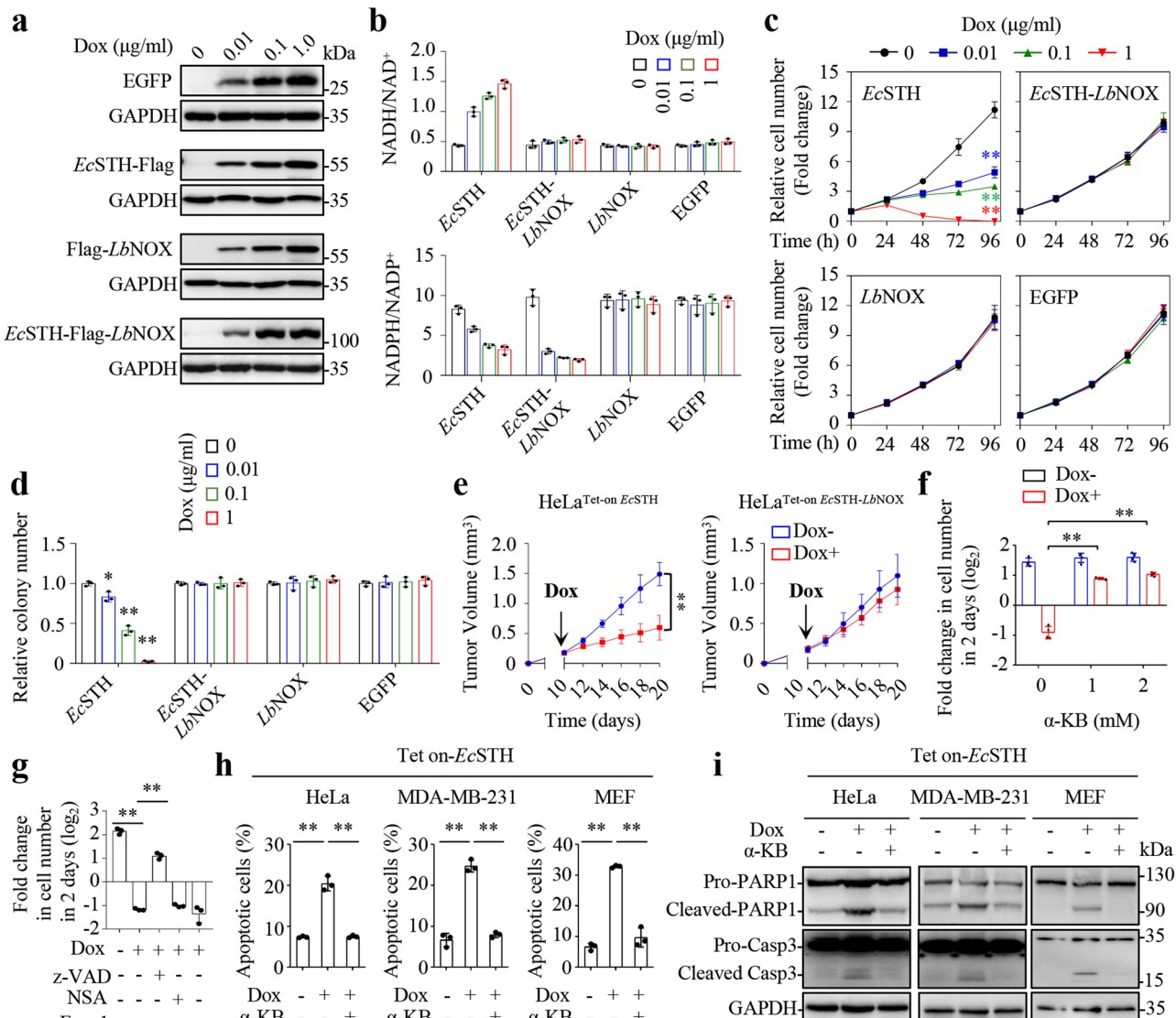

**Fig. 3 | EcSTH causes growth inhibition and cell death by inducing NADH accumulation. a** Western blots for the expression of inducible Tet-on EGFP, *Ec*STH, *Lb*NOX or *Ec*STH-*Lb*NOX in HeLa cells induced by Dox for 12 h. **b** The ratios of cellular NADH/NAD⁺ and NADPH/NADP⁺ in HeLa cells expressing Tet-on EGFP, *Ec*STH, *Lb*NOX, or *Ec*STH-*Lb*NOX cultured with different concentrations of Dox for 12 h (*n* = 3 biologically independent samples). **c** Effects of the expression of EGFP, *Ec*STH, *Lb*NOX or *Ec*STH-*Lb*NOX on the proliferation of HeLa cells cultured with different concentrations of Dox as indicated (*n* = 3 biologically independent samples). From top to bottom: **P < 0.0001, **P < 0.0001, **P < 0.0001, two-way ANOVA. **d** Effects of EGFP, *Ec*STH, *Lb*NOX, or *Ec*STH-*Lb*NOX expression on colony formation in HeLa cells cultured with Dox, as indicated, for 10 days (*n* = 3 biologically independent samples). From left to right: *P = 0.013, **P = 0.0001, **P = 6.58e-07, two-tailed Student's *t*-tests. **e** The in vivo tumor growth of HeLa cells expressing inducible Tet-on *Ec*STH or *Ec*STH-*Lb*NOX (*n* = 5 biologically independent animals). 2 mg/mL Dox was added in drinking water on day 10 and updated every two days. **P < 0.0001, two-way ANOVA. **f** Effects of α-KB treatment on proliferation of

HeLa^Tet-on *Ec*STH cells. Cells were counted after treatment with Dox (1 μg/mL) and α-KB (1 or 2 mM) for 48 h (*n* = 4 biologically independent samples). From left to right: **P = 2.44e-06, **P = 1.85e-06, two-tailed Student's *t*-tests. **g** Effects of inhibitors on proliferation of HeLa^Tet-on *Ec*STH cells. Cells were pretreated with the indicated inhibitors for 2 h, and cells were counted after treatment with Dox (1 μg/mL) for 48 h (*n* = 3 biologically independent samples). z-VAD (z-VAD-FMK, 20 μM), NSA (necrosulfonamide, 10 μM), Ferr-1 (Ferrostain-1, 5 μM). From left to right: **P = 1.74e-06, **P = 9.64e-06, two-tailed Student's *t*-tests. **h** Effects of α-KB treatment on cell apoptosis. Cell apoptosis was measured by Flow cytometry after treatment with Dox (1 μg/mL) and α-KB (2 mM) for 24 h (*n* = 3 biologically independent samples). From left to right: **P = 0.00023, **P = 0.00024, **P = 0.00013, **P = 5.51e-05, **P = 3.22e-06, **P = 0.00021, two-tailed Student's *t*-tests. **i** Western blots for the cleavage of PARP-1 and Caspase 3. Cells were treated with Dox (1 μg/mL) and α-KB (2 mM) for 24 h. All experimental data were verified in at least three independent experiments. Error bars represent mean ± SD. *P < 0.05, **P < 0.01. Source data are provided as a Source Data file.

and MEF cells induced by *Ec*STH expression (Fig. 3h, i; Supplementary Figs. 4f and 5f).

Intriguingly, our results showed that *Ec*STH-*Lb*NOX, *Ec*STH or *Lb*NOX did not affect the ratios or contents of GSH and/or GSSG (Supplementary Fig. 6a). An oxidative inducer, menadione, potently killed cells, but such an effect was largely reversed by the addition of GSH monoethyl ester or NAC (N-acetyl-L-cysteine, a precursor of GSH) (Supplementary Fig. 6b). In sharp contrast, supplementation with GSH

monoethyl ester or NAC did not rescue cell death induced by *Ec*STH at all (Supplementary Fig. 6b). Meanwhile, we also found that cells with or without expression of *Ec*STH or *Ec*STH-*Lb*NOX had the similar sensitivity to menadione (Supplementary Fig. 6c). These data suggested that oxidative stress was not involved in *Ec*STH-induced cell death. Although *Lb*NOX fusion almost completely restored the cellular level of NADH and the ratio of NADH/NAD⁺, it did not reverse the decreased ratio of NADPH/NADP⁺, with cellular NADPH being reduced

to no less than 60% (Fig. 3b and Supplementary Figs. 3b, 4c and 5c). However, *Ec*STH-*Lb*NOX did not influence cell growth in vivo and in vitro (Fig. 3c, e; Supplementary Figs. 4d and 5d). Therefore, it appears that most likely, cellular NADPH or NADPH/NADP⁺, even declined to some extent, supports cell proliferation and survival sufficiently. These data suggest that NADH accumulation plays a critical role in *Ec*STH-induced cell growth arrest or cell death. Taken together, *Ec*STH can function as a genetic tool in cell lines to simulate NADH-accumulation alone or in combination with *Lb*NOX.

### Elevated purine biosynthesis is associated with NADH accumulation-induced cell death

Our results have indicated that *Ec*STH expression repressed cell proliferation upon induction by 0.1 µg/mL Dox, while its expression induced by 1 µg/mL Dox killed cells. Importantly, the inhibitory or cell death effects of *Ec*STH expression were dependent on NADH accumulation. Therefore, to determine the NADH accumulation-sensing pathway associated with cell death, we investigated whether knocking out some unknown gene(s) could exacerbate this pathway to elicit cell death under the expression of *Ec*STH induced by 0.1 µg/mL Dox (Fig. 4a). We infected the cells expressing Cas9 and inducible *Ec*STH with two human genome-wide CRISPR/Cas9-based lentiviral sgRNA libraries[19], GeCKOV2 and[20] Brunello, respectively (Fig. 4b). For the GeCKOV2 library, cells were cultured for three days without Dox, or for seven days with 0.1 µg/mL Dox, so that they had similar doubling numbers, with the cell number under both conditions being increased by about eight times. Cells infected with the Brunello library were cultured for nine days with or without 0.1 µg/mL Dox. Eventually, the same number of cells underwent deep sequencing to identify the loss of genes based on the negative scores (Supplementary Data 1). Among the top 10 scores, only one overlapping gene, transketolase (TKT), was identified (Fig. 4b and Supplementary Fig. 7a). TKT is a key enzyme in the non-oxidative pentose phosphate pathway (PPP) and it connects oxidative PPP with glycolysis and determines whether ribose 5-phosphate feeds into glycolysis or purine biosynthesis (Fig. 4c). We then confirmed that induction by 0.1 µg/mL Dox, *Ec*STH, but not *Ec*STH-*Lb*NOX, dramatically killed TKT-knockout HeLa and MDA-MB-231 cells using pooled sgRNA-derived lentivirus (Fig. 4d and Supplementary Fig. 7b).

We now performed metabolic profiling on HeLa/sgControl and HeLa/sgTKT cells, and found that TKT-knockout increased the levels of cellular metabolites in the pathways of glycolysis, PPP and purine biosynthesis (Supplementary Fig. 7c). However, *Ec*STH expression did not affect the protein level or activity of TKT (Supplementary Fig. 7d), and exogenous over-expression of TKT did not prevent cell growth arrest or cell death induced by *Ec*STH expression (Supplementary Fig. 7e). Therefore, to determine how TKT blockade exacerbated cell death induced by *Ec*STH expression, we carried out a targeted metabolomics analysis on cells expressing *Ec*STH with or without TKT-knockout. The principal component analysis of the raw data showed that the samples from *Ec*STH-expressing HeLa cells/sgTKT were sharply separated from those of *Ec*STH-expressing cells/sgControl (Fig. 4e) and the decreased metabolites were scattered in many pathways (Supplementary Fig. 7f, g). By contrast, increased metabolites caused by TKT-knockout were significantly enriched in purine biosynthesis and PPP (Fig. 4f, g; Supplementary Fig. 7g). We next tested whether TKT-knockout forced ribose 5-phosphate, produced in the oxidative PPP that could be promoted by the NADPH consumption, into purine biosynthesis to accelerate cell death. We treated cells with 6-mercaptopurine (6-MP), whose secondary metabolite potently blocks amidophosphoribosyl transferase (PPAT) catabolizing the first step of purine biosynthesis[21,22], and the two inhibitors, pelitrexol and lometrexol, of trifunctional purine biosynthetic protein adenosine-3 (GART) involved in purine biosynthesis[23,24]. Our results showed that 6-MP, pelitrexol and lometrexol significantly suppressed death in HeLa/

Tet on-*Ec*STH and MDA-MB-231/Tet on-*Ec*STH cells induced by TKT-knockout in the presence of 0.1 µg/mL Dox (Fig. 4h). Therefore, elevated purine biosynthesis is tightly associated with NADH accumulation-induced cell death.

### Purine biosynthesis functions as an NADH accumulation-sensing pathway

To determine how NADH accumulation affects purine biosynthesis, we next performed metabolomic profiling on HeLa cells expressing *Ec*STH, *Lb*NOX or *Ec*STH-*Lb*NOX (Supplementary Data 2). Compared with *Ec*STH-*Lb*NOX, *Ec*STH highly enriched the cellular metabolites involved in nucleotide biosynthesis, and glycolysis (Fig. 5a). Moreover, all these metabolites were largely labeled by ¹³C₆-glucose (Supplementary Fig. 8), indicating the presence of accelerated metabolic transformations in cells expressing *Ec*STH. *Lb*NOX itself showed no or slight effects on the cellular metabolites, whereas *Ec*STH-induced nucleotides, as well as glycolytic intermediates, were significantly blocked in cells with *Ec*STH-*Lb*NOX expression (Supplementary Fig. 9a, b). Notably, *Ec*STH reduced the cellular levels of tricarboxylic acid (TCA) cycle metabolites (Supplementary Fig. 10). However, *Ec*STH-*Lb*NOX fusion did not reverse the decreased TCA cycle metabolites (Supplementary Fig. 10). Therefore, the decreased TCA cycle metabolites seemed to be unrelated to cell growth in these cells. To further investigate the role of *Ec*STH expression in the promotion of de novo purine biosynthesis, we used ¹⁵N-amide-glutamine to trace its incorporation. ¹⁵N-amide-glutamine can label IMP and AMP with two ¹⁵N atoms (m + 2), and GMP with three ¹⁵N atoms (m + 3) (Fig. 5b). Our results showed that *Ec*STH expression remarkably enhanced the cellular levels of IMP m + 2, AMP m + 2 and GMP m + 3, and the total contents, and this effect was completely abolished in cells expressing *Ec*STH-*Lb*NOX (Fig. 5c and Supplementary Fig. 11a). These results demonstrate that *Ec*STH-induced NADH accumulation promotes de novo purine biosynthesis.

Furthermore, our results demonstrated that antimycin A and hypoxia increased the cellular levels of de novo synthesized purine nucleotides, as indicated by the labeling of IMP, AMP and GMP by ¹⁵N-amide-glutamine in HeLa and MDA-MB-231 cells (Supplementary Fig. 11b–e). Next, to determine whether NADH accumulation exerted a critical role in de novo purine biosynthesis under hypoxia and ETC inhibition, we supplemented cells with α-ketobutyrate to attenuate NADH accumulation. Our results showed that α-ketobutyrate significantly reduced the levels of ¹⁵N-labeled and total IMP, AMP and GMP in HeLa and MDA-MB-231 cells under hypoxia or ETC inhibition (Supplementary Fig. 11b–e), again supporting the contribution of NADH accumulation to purine biosynthesis.

6-MP, pelitrexol and lometrexol dramatically blocked purine biosynthesis and reduced the ¹⁵N-glutamine-labeled and total level of purines in *Ec*STH-expressing cells (Supplementary Fig. 12a, b). Meanwhile, these inhibitors also significantly prevented death in HeLa, MDA-MB-231 and MEF cells expressing *Ec*STH induced by 1 µg/mL Dox (Fig. 5d and Supplementary Fig. 12c, d). Taken together, our data suggest that purine biosynthesis can function as a downstream pathway with the ability to sense accumulated NADH.

### NADH indirectly activates PRPS2

We next attempted to determine how NADH accumulation triggered purine biosynthesis. We noticed that although *Ec*STH-*Lb*NOX expression, compared with *Ec*STH, further accumulated cellular ribose 5-phosphate, it did not increase the level of PRPP (phosphoribosyl pyrophosphate) (Fig. 5e), as well as the downstream nucleotides (Supplementary Fig. 9a, b). These data suggest that the elevated cellular ribose 5-phosphate does not promote the biosynthesis of PRPP and its downstream nucleotides, if cellular NADH is not accumulated. PRPS1/2 was responsible for the conversion of ribose 5-phosphate into PRPP, thus could be one critical sensor for the detection of the accumulated

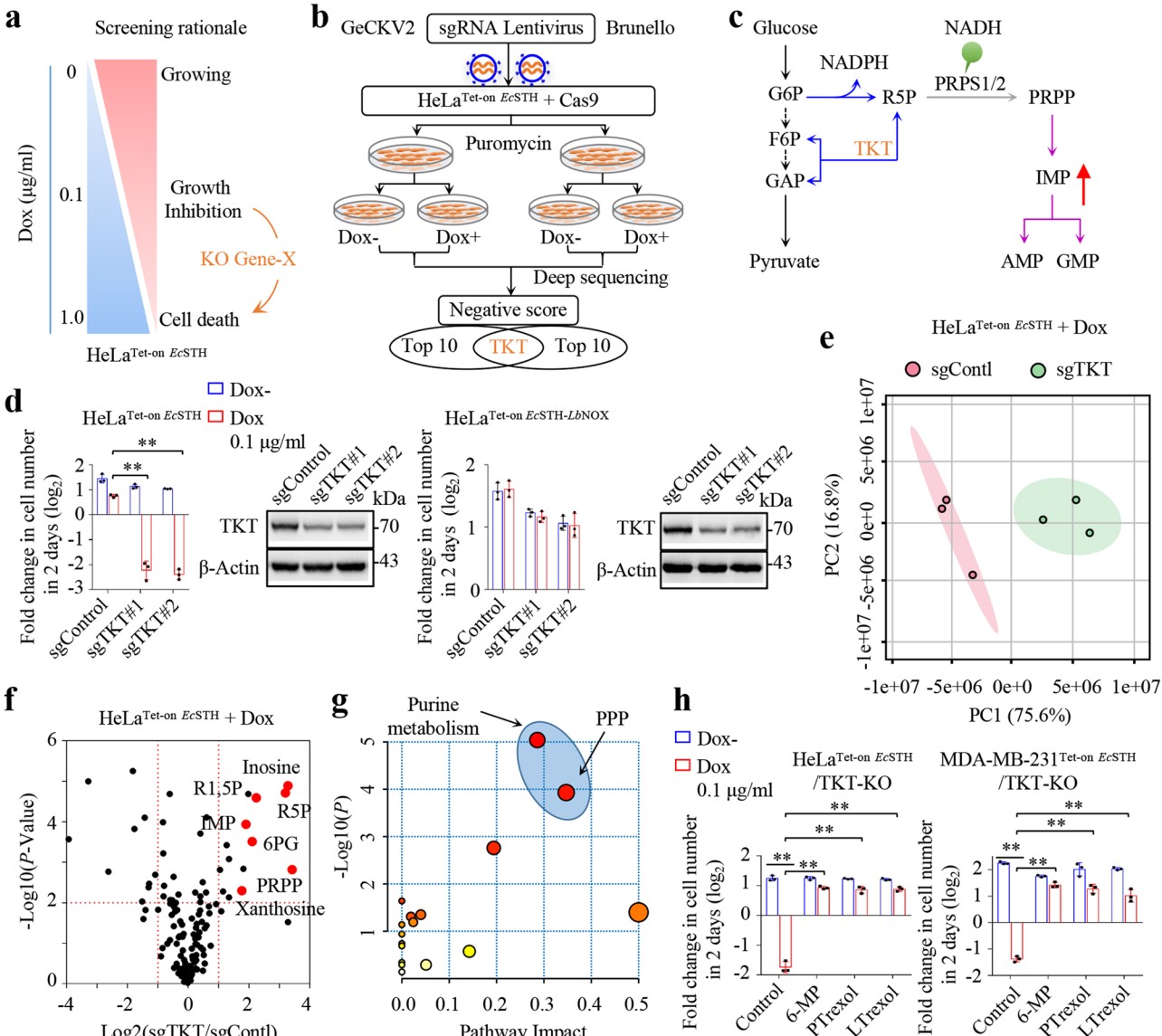

**Fig. 4 | Deregulated purine biosynthesis is associated with NADH accumulation-induced cell death. a** The rationale of screening for NADH accumulation-associated cell death-promoting genes. **b** Workflow for the CRISPR/Cas9 screening strategy. Two sgRNA pools (Human GeCKO v2 and Brunello libraries) were used. **c** Metabolic processes coupling glycolysis (black lines), pentose phosphate pathway (blue lines), and purine biosynthesis (purple lines). The green balloon shows the potential activation of PRPS by NADH. G6P, glucose 6-phosphate; F6P, fructose 6-phosphate; GAP, glyceraldehyde 3-phosphate; R5P, Ribose 5-phosphate; PRPP, phosphoribosyl pyrophosphate; IMP, inosine monophosphate; AMP, adenosine monophosphate; GMP, guanosine monophosphate; **d** Effects of TKT knockout on the proliferation of HeLa$^{Tet-on\ EcSTH}$ or HeLa$^{Tet-on\ EcSTH-LbNOX}$ cells ($n = 3$ biologically independent samples). Cells were counted after treatment with Dox (0.1 μg/mL) for 48 h. The immunoblot showed the levels of TKT. From left to right: **$P = 0.00015$, **$P = 1.48e-05$. **e** Principal-component analysis of metabolites in HeLa$^{Tet-on\ EcSTH}$ cells with sgTKT versus sgControl cultured with Dox (0.1 μg/mL)

for 24 h ($n = 3$ biologically independent samples). **f** Volcano plot of the differential metabolites (two-tailed Student's $t$-tests) between sgTKT and sgControl in HeLa$^{Tet-on\ EcSTH}$ cells cultured with Dox (0.1 μg/mL) for 24 h ($n = 3$ biologically independent samples). The red highlights represent metabolites involved in purine metabolism and the pentose phosphate pathway. **g** KEGG pathway analysis of the upregulated metabolites (fold change > 1.5, $P$ value < 0.05) using MetaboAnalyst 5.0. **h** Effects of purine biosynthesis inhibitors on proliferation of TKT knockout HeLa$^{Tet-on\ EcSTH}$ and MDA-MB-231$^{Tet-on\ EcSTH}$ cells ($n = 3$ biologically independent samples). Cells were pretreated with the indicated inhibitors for 2 h and counted after treatment with Dox (0.1 μg/mL) and inhibitors for 48 h. 6-Mercaptopurine (6-MP, 300 μM), pelitrexol (PTrexol, 10 μM), lometrexol (LTrexol, 10 μM). From left to right: **$P = 1.53e-05$, **$P = 1.85e-05$, **$P = 3.40e-05$, **$P = 2.31e-05$, **$P = 7.37e-07$, **$P = 5.72e-06$, **$P = 2.24e-05$, **$P = 8.99e-05$. **d, h** Data were verified in at least three independent experiments. Error bars represent mean ± SD. **$P < 0.01$; Two-tailed Student's $t$-tests. Source data are provided as a Source Data file.

NADH. Our results showed that the expression of *Ec*STH or *Ec*STH-*Lb*NOX did not affect the protein levels of PRPS1 and PRPS2 (Supplementary Fig. 13a). Therefore, to determine whether PRPS1 or PRPS2 played an essential role, we used siRNAs to suppress the expression of PRPS1 and/or PRPS2 (Fig. 5f). The depletion of PRPS1 and/or PRPS2 did not affect the increased NADH/NAD⁺ ratio (Fig. 5g), but they (in particular PRPS2 knockdown) significantly prevented cell death induced

by *Ec*STH (Fig. 5h). Next, we purified recombinant human PRPS1 and PRPS2 in the bacterial system and tested the effects of NADH and NAD⁺ on their activities. Intriguingly, both NADH and NAD⁺ did not affect the activity at all (Supplementary Fig. 13b). Considering that PRPS1/2 activity was mainly regulated by physiological allosteric effectors, such as ADP and GDP[25], we then sought to investigate whether NADH and NAD⁺ influenced such regulations. Our results showed that NADH and

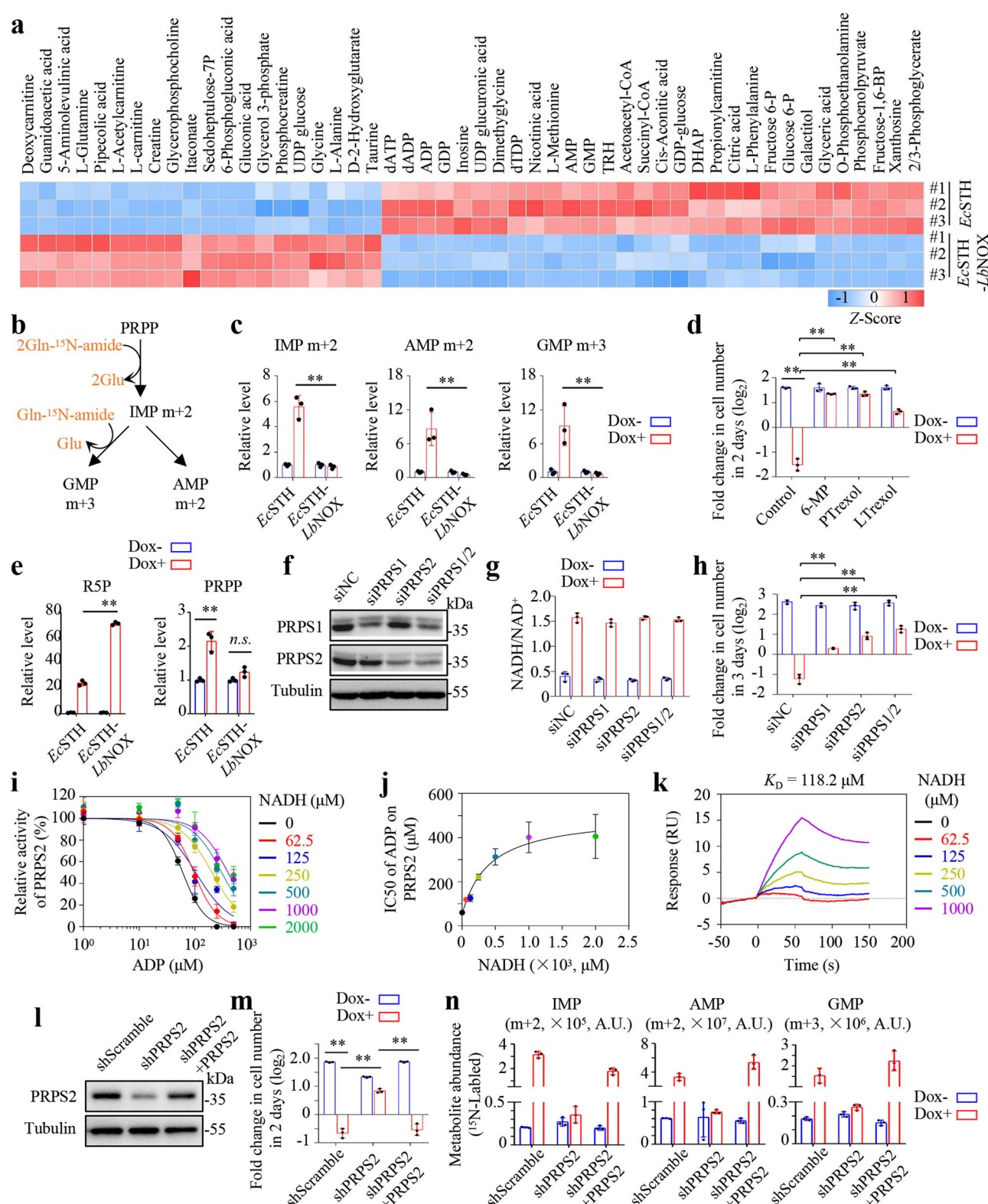

NAD[+] exerted no effect on the suppression of PRPS1 by ADP or GDP (Supplementary Fig. 13c). Interestingly, NADH, but not NAD[+], significantly protected PRPS2 against inhibition by ADP, but not against that by GDP (Supplementary Fig. 13d). ADP exhibited a much more potent inhibitory effect on PRPS1/2 than GDP (Supplementary Fig. 13c, d), and represents the major physiological inhibitor[26]. NADH antagonized the inhibitory effect of ADP on PRPS2 in a concentration-dependent manner (Fig. 5i), enhancing the IC50 value of ADP by about

six times up to its maximum level (Fig. 5j). Our results further demonstrated that NADH bound to PRPS2 with a dissociation constant of 118.2 μM, as revealed by Biacore analysis (Fig. 5k), which is comparable to the cytosolic concentration, ranging around 100 μM, of NADH in mammalian cells[27], rendering NADH as a suitable physiological regulator of PRPS2. We further stably knocked down PRPS2 in HeLa[EcSTH] cells and also re-expressed PRPS2 to restore it in knockdown cells (Fig. 5l). As expected, EcSTH expression induced NADH

**Fig. 5 | Purine biosynthesis functions as an NADH accumulation-sensing pathway. a** Heat map showing the top 50 differential metabolites between HeLa$^{Tet-on\ EcSTH}$ and HeLa$^{Tet-on\ EcSTH-LbNOX}$ cells cultured with Dox (0.1 μg/mL) for 24 h ($n = 3$ biologically independent samples). Z-score was calculated according to their fold changes (The abundance of main metabolites was listed as Supplementary Fig. 9a, b). TRH, thyrotropin-releasing hormone; DHAP, dihydroxyacetone phosphate. **b** Schematic diagram illustrating the labeling of de novo purine biosynthesis by glutamine-$^{15}$N-amide nitrogen. **c** The $^{15}$N-labeled fraction of IMP, AMP and GMP in HeLa$^{Tet-on\ EcSTH}$ and HeLa$^{Tet-on\ EcSTH-LbNOX}$ cells cultured with glutamine-$^{15}$N-amide (1 mM) for 12 h ($n = 3$ biologically independent samples). Cells were pretreated with Dox (0.1 μg/mL) for 12 h. From left to right: **$P = 0.00087$, **$P = 0.00933$, *$P = 0.01399$. **d** Effects of purine biosynthesis inhibitors on proliferation of HeLa$^{Tet-on\ EcSTH}$ cells ($n = 3$ biologically independent samples). Cells were pretreated with the indicated inhibitors for 2 h and counted after treatment with Dox (1 μg/mL) and inhibitors for 48 h. 6-Mercaptopurine (6-MP, 300 μM), pelitrexol (PTrexol, 10 μM), lometrexol (LTrexol, 10 μM). From left to right: **$P = 2.35e-05$, **$P = 3.21e-05$, **$P = 4.30e-05$, *$P = 0.00013$. **e** Intracellular ribose 5-phosphate (R5P) and phosphoribosyl pyrophosphate (PRPP) levels determined by LC-MS in HeLa$^{Tet-on\ EcSTH}$ and HeLa$^{Tet-on\ EcSTH-LbNOX}$ cells cultured with Dox (0.1 μg/mL) for 24 h ($n = 3$ biologically independent samples). From left to right: **$P = 4.03e-06$, **$P = 0.00248$, ns

$P = 0.05472$. **f** Immunoblots showing the levels of PRPS1, PRPS2, and α-Tubulin. siPRPS1 or siPRPS2 pool was mixed with three independent siRNAs. **g** Effect of PRPS1/2 knockdown on NADH/NAD$^+$ in HeLa$^{Tet-on\ EcSTH}$ cells cultured with Dox (1 μg/mL) for 12 h ($n = 3$ biologically independent samples). **h** Effects of PRPS1/2 knockdown on proliferation of HeLa$^{Tet-on\ EcSTH}$ cells ($n = 3$ biologically independent samples). From left to right: **$P = 1.34e-05$, **$P = 0.00039$, **$P = 0.00023$, *$P = 0.00012$. **i** Effects of NADH on PRPS2 activity in the presence of ADP as indicated ($n = 3$ biologically independent samples). **j** The IC50 value of ADP against PRPS2 activity as calculated under different NADH concentrations based on data from **i**. **k** The affinity of NADH to PRPS2 detected by SPR. **l** Immunoblots showing the levels of PRPS2 and α-Tubulin. **m** Effects of PRPS2 knockdown as well as PRPS2 rescue on the proliferation of HeLa$^{Tet-on\ EcSTH}$ cells ($n = 3$ biologically independent samples). From left to right: **$P = 1.32e-05$, **$P = 0.00014$, **$P = 0.00038$. **n** The $^{15}$N-labeled fraction of IMP, AMP, and GMP in PRPS2 knockdown as well as PRPS2 rescue HeLa$^{Tet-on\ EcSTH}$ cells cultured with glutamine-$^{15}$N-amide (1 mM) for 12 h ($n = 3$ biologically independent samples). Cells were pretreated with Dox (0.1 μg/ mL) for 12 h. **d, f–m** Data were verified in at least three independent experiments. Error bars represent mean ± SD. **$P < 0.01$; Two-tailed Student's $t$-tests. Source data are provided as a Source Data file.

accumulation regardless of the states of PRPS2 (Supplementary Fig. 13e), but it only drastically promoted the cellular levels of newly-synthesized and total IMP, AMP and GMP, and elicited death in the PRPS2-expressing cells while not in the knockdown cells (Fig. 5m, n; Supplementary Fig. 13f). Interestingly, under the normal condition, PRPS2 knockdown did not affect the biosynthesis of IMP, AMP and GMP, but almost completely prevented EcSTH-promoted purine biosynthesis (Fig. 5n). Therefore, the elevated NADH level can potentially activate PRPS2 by relieving it from the physiological feedback inhibition afforded by its end product.

We next tried to conduct a molecular docking analysis to identify the binding sites of NADH and ADP in the predicted structure of PRPS2 by Alphafold from UniProt. The docking results showed that the NADH-binding site and ADP-binding site in PRPS2 were indeed overlapped (Supplementary Fig. 14a–c). In addition, we also docked NADH and ADP on PRPS1 using its crystal structure, and found that the ADP-binding site of PRPS1 shared critical amino acid residues with that of PRPS2. Although an NADH-binding site was also predicted in PRPS1, it was very different from the ADP-binding site (Supplementary Fig. 14d–f). However, all the amino acid residues contributing to the binding of ADP and NADH to PRPS1 or PRPS2 are among the identical residues of both PRPS1 and PRPS2. Meanwhile, based on the structures, we noticed that all the different amino acid residues were sporadically distributed on the surface of proteins. Therefore, they most likely affected the binding of NADH to PRPS2 by subtly changing the structure.

## Purine biosynthesis contributes to energy stress-mediated cell death induced by NADH accumulation

Purine biosynthesis itself requires immense ATP investment, consuming 10 ATP and 9 ATP molecules for de novo biosynthesis of GMP and AMP, respectively[6]. Therefore, the deregulated purine biosynthesis could provoke massive energy consumption directly in cells. As a canonical indicator of energy stress, AMPK phosphorylation at Thr172, and the phosphorylation of its substrate, acetyl-coenzyme A carboxylase (ACC), were detected in cells expressing EcSTH but not in those expressing EcSTH-LbNOX (Supplementary Fig. 15a). We indeed observed that EcSTH expression significantly decreased cellular ATP levels, which were completely reversed in EcSTH-LbNOX-expressing cells (Fig. 6a). In contrast, EGFP and LbNOX did not affect cellular ATP (Fig. 6a). α-Ketobutyrate, which attenuated NADH accumulation (Supplementary Fig. 1b) and enhanced cell survival (Fig. 3f, h), also significantly restored cellular ATP level in EcSTH-expressing cells (Fig. 6b). Knockdown of PRPS2, or inhibitors of purine biosynthesis, 6-MP, pelitrexol and lometrexol, also significantly restored cellular ATP

in cells expressing EcSTH (Fig. 6c, d). Correspondingly, these inhibitors significantly suppressed the phosphorylation of AMPK and ACC1 induced by EcSTH (Fig. 6e), indicating that the increased purine biosynthesis is the major cause of the energy stress induced by NADH accumulation. Notably, the intermediate, AICAR, involved in the purine biosynthetic pathway could potentially activate AMPK, and it was also promoted by EcSTH and suppressed by the purine biosynthesis inhibitors (Supplementary Fig. 15b). Therefore, to determine whether energy stress played a critical role in EcSTH-induced cell death, we pre-activated AMPK in cells with its activators, AICAR and A769662 (Supplementary Fig. 15c), to attenuate energy stress. We indeed observed that the two activators significantly prevented ATP loss (Fig. 6f), and blocked cell death induced by EcSTH (Fig. 6g and Supplementary Fig. 15d). These results suggest that energy stress is the cause of EcSTH-induced cell death, while the concomitant AMPK activation possibly only functions as the negative feedback regulator. In addition, inhibition of purine biosynthesis or activation of AMPK did not affect NADH accumulation (Supplementary Fig. 15e). Therefore, these data indicate that the accumulated NADH is an apical event to initiate purine biosynthesis-driven energy crisis as the inducer of cell death (Supplementary Fig. 15f).

## Blocking purine biosynthesis prevents liver damage by ethanol administration-induced NADH accumulation

Active oxidation of some metabolites can lead to NADH accumulation. For example, ethanol can be oxidized to acetate by alcohol dehydrogenase and aldehyde dehydrogenase, concomitantly producing and accumulating NADH to induce reductive stress[3]. To test whether the metabolic NADH accumulation exerted a critical pathophysiological role associated with its promotion of purine biosynthesis in vivo, we intragastrically administrated ethanol to mice (Fig. 7a). Our results showed that ethanol administration significantly increased the level of NADH and ratio of NADH/NAD$^+$ while reducing the content of NADPH and ratio of NADPH/NADP$^+$ in liver tissue (Fig. 7b and Supplementary Fig. 16a, b). Meanwhile, ethanol administration enhanced the levels of IMP, AMP and GMP, whereas decreasing the ATP content in liver tissue (Fig. 7c, d). We further detected a dramatic increase in blood levels of alanine aminotransferase (ALT) and aspartate aminotransferase (AST), the widely used indicators of liver damage, in mice given ethanol (Fig. 7e). Therefore, we further measured liver injury using terminal deoxynucleotidyl transferase dUTP nick end labeling (TUNEL) to detect DNA fragmentation in dead cells. Using a FITC-dUTP TUNEL assay on the paraffin sections, our results showed that ethanol administration resulted in obvious FITC-labeling (Fig. 7f, g). Next, we investigated

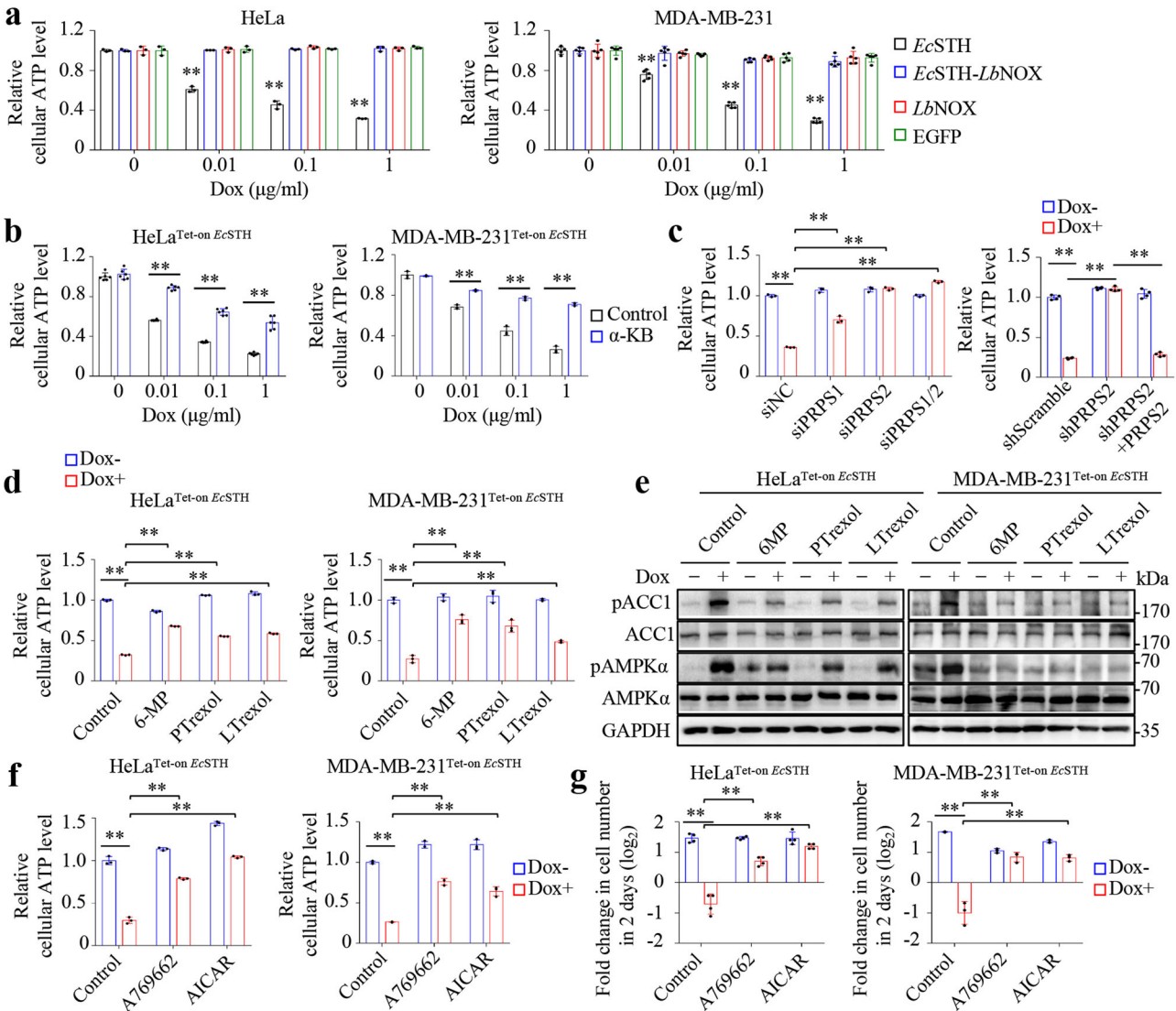

**Fig. 6 | NADH accumulation activates purine biosynthesis to induce energy crisis. a** Cellular ATP levels of HeLa ($n = 3$ biologically independent samples) and MDA-MB-231 cells ($n = 5$ biologically independent samples) expressing Tet-on EGFP, $Ec$STH, $Lb$NOX, or $Ec$STH-$Lb$NOX cultured with Dox as indicated for 12 h. From left to right: **$P = 2.44$e-05, **$P = 2.34$e-05, **$P = 6.13$e-08, **$P = 0.00277$, **$P = 8.62$e-05, **$P = 2.55$e-05. **b** Cellular ATP level of HeLa$^{Tet-on EcSTH}$ ($n = 6$ biologically independent samples) and MDA-MB-231$^{Tet-on EcSTH}$ cells ($n = 3$ biologically independent samples) cultured with Dox and α-KB (2 mM) as indicated for 12 h. From left to right: **$P = 1.56$e-11, **$P = 7.57$e-10, **$P = 3.68$e-07, **$P = 0.00020$, **$P = 0.00030$, **$P = 2.73$e-05. **c**, Effects of PRPS1/2 knockdown on cellular ATP level in HeLa$^{Tet-on EcSTH}$ cells cultured with Dox (1 μg/mL) for 12 h ($n = 3$ biologically independent samples for siPRPS2 and $n = 4$ biologically independent samples for shPRPS2). From left to right: **$P = 2.45$e-07, **$P = 0.00015$, **$P = 3.82$e-07, **$P = 6.59$e-08, **$P = 5.67$e-09, **$P = 2.06$e-09, **$P = 1.12$e-08. **d** Cellular ATP level of HeLa$^{Tet-on EcSTH}$ and MDA-MB-231$^{Tet-on EcSTH}$ cells cultured with Dox and purine synthesis inhibitors as indicated for 12 h ($n = 3$ biologically independent samples). From left to right: **$P = 6.90$e-08, **$P = 6.46$e-08, **$P = 2.86$e-07, **$P = 9.23$e-07,

**$P = 2.67$e-05, **$P = 0.00028$, **$P = 0.00125$, **$P = 0.00150$. **e** Effects of purine synthesis inhibitors on AMPK pathway in HeLa$^{Tet-on EcSTH}$ and MDA-MB-231$^{Tet-on EcSTH}$ cells cultured with Dox (1 μg/mL) and inhibitors for 48 h. **f** Cellular ATP level of HeLa$^{Tet-on EcSTH}$ and MDA-MB-231$^{Tet-on EcSTH}$ cells cultured with Dox and AMPK activators as indicated for 12 h ($n = 3$ biologically independent samples). From left to right: **$P = 3.52$e-05, **$P = 2.63$e-05, **$P = 5.27$e-06, **$P = 1.90$e-07, **$P = 2.87$e-05, **$P = 0.00033$. **g** Effects of AMPK activators on the proliferation of HeLa$^{Tet-on EcSTH}$ ($n = 4$ biologically independent samples) and MDA-MB-231$^{Tet-on EcSTH}$ ($n = 3$ biologically independent samples) cells. Cells were counted after treatment with Dox (1 μg/mL) and activators as indicated for 48 h. From left to right: **$P = 2.02$e-05, **$P = 0.00126$, **$P = 0.00064$, **$P = 0.00025$, **$P = 0.00145$, **$P = 0.00129$. **d–g** Cells were pretreated with the indicated inhibitors or activators for 2 h. 6-MP (300 μM), PTrexol (10 μM), LTrexol (10 μM), A769662 (100 μM), and AICAR (400 μM) were used. All experimental data were verified in at least three independent experiments. Error bars represent mean ± SD. **$P < 0.01$; Two-tailed Student's $t$-tests. Source data are provided as a Source Data file.

whether NADH accumulation initiated downstream events, therefore, we injected $Lb$NOX-expressing lentivirus into the caudal vein of mice three days before ethanol administration (Fig. 7a). Intravenous viruses are predominantly trapped in liver tissue[3], and here we also confirmed the expression of $Lb$NOX in the liver (Fig. 7a). We observed that $Lb$NOX indeed restored the ratios of NADH/NAD$^+$ and NADPH/NADP$^+$ (Fig. 7b), and almost completely scavenged the above phenotypes associated with ethanol administration (Fig. 6c–g). By

contrast, pretreatment of 6-MP or pelitrexol did not affect the homeostasis of NADH/NAD$^+$ and NADPH/NADP$^+$ (Fig. 7b; Supplementary Fig. 16a, b), but significantly suppressed the levels of IMP, AMP and GMP (Fig. 7c), restored ATP content (Fig. 7d) and meantime prevented tissue injury (Fig. 7e–g) in the liver tissue of mice given ethanol. Taken together, our data suggest that blocking purine biosynthesis can potentially protect against tissue damage induced by metabolically reductive stress.

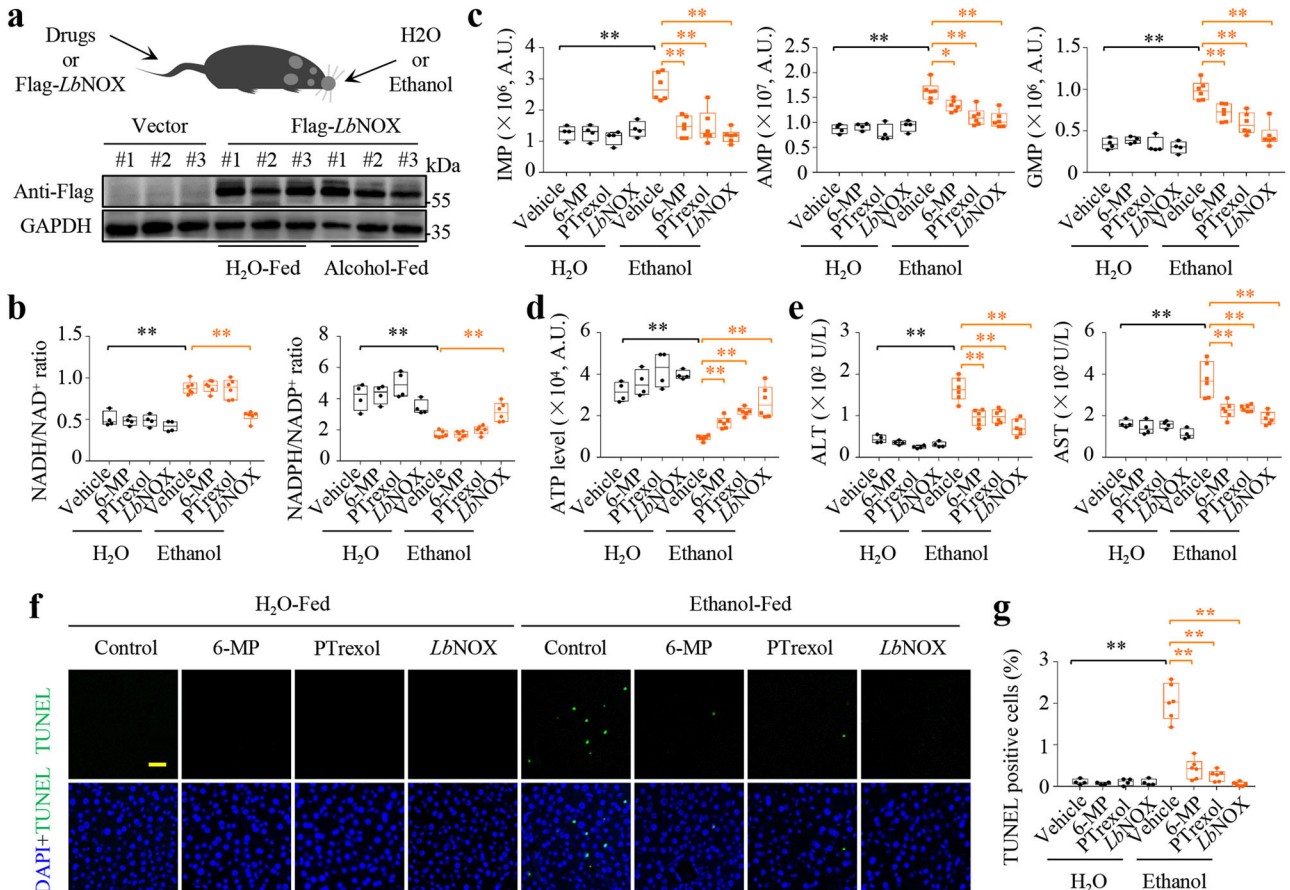

**Fig. 7 | Blocking purine biosynthesis prevents liver damage by ethanol administration-induced reductive stress. a** Schematic diagram illustrating the acute ethanol gavage model (top) and lentivirus-mediated *Lb*NOX expression in liver tissue (bottom). 6-MP (100 mg/kg) and PTrexol (10 mg/kg) were pre-injected into the tail vein for 1 h and lentivirus ($3*10^8$ PFU) were pre-injected into the tail vein for 3 days. Mice were sacrificed after ethanol (6 g/kg) gavage for 9 h. **b** Effects of acute ethanol gavage on the ratio of NADH/NAD$^+$ and NADPH/NADP$^+$. From left to right: **$P$ = 0.00012, **$P$ = 8.67e-06, **$P$ = 0.00012, **$P$ = 0.00026. **c** Effects of acute ethanol gavage on liver tissue IMP, AMP, and GMP levels detected by LC/MS. From left to right: for IMP, **$P$ = 0.00029, **$P$ = 0.00020, **$P$ = 0.00103, **$P$ = 1.29e-05; for AMP, **$P$ = 8.41e-05, *$P$ = 0.01049, **$P$ = 0.00064, **$P$ = 0.00023; for GMP, **$P$ = 8.11e-06, **$P$ = 0.00165, **$P$ = 0.00021, **$P$ = 2.15e-05. **d** Effects of acute ethanol

gavage on liver tissue ATP levels. From left to right: **$P$ = 3.58e-06, **$P$ = 0.00011, **$P$ = 8.46e-08, **$P$ = 0.00027. **e** Serum alanine aminotransferase (ALT) and aspartate aminotransferase (AST) activity in ethanol gavage groups versus control groups. From left to right: for ALT, **$P$ = 4.03e-05, **$P$ = 0.00055, **$P$ = 0.00062, **$P$ = 5.78e-05; for AST, **$P$ = 0.00111, *$P$ = 0.00277, **$P$ = 0.00248, **$P$ = 0.00042. **f, g** Imaging **f** and quantitation **g** of liver cell death detected by TUNEL staining after acute ethanol gavage for 9 h. Scale bar, 25 µm. From left to right: **$P$ = 2.56e-05, **$P$ = 1.22e-05, **$P$ = 2.41e-06, **$P$ = 6.21e-07. H$_2$O group ($n$ = 4 biologically independent animals) and ethanol group ($n$ = 6 biologically independent animals). **c**–**e, g** Boxes represent minima to maxima (line at median) with whiskers at 1.5*IQR. *$P$ < 0.05, **$P$ < 0.01; Two-tailed Student's $t$-tests. Source data are provided as a Source Data file.

## Discussion

Although reductive stress features an increased NADH/NAD$^+$ ratio (NADH accumulation), it seems to be tightly associated with a decreased NADPH/NADP$^+$ ratio. However, most studies on reductive stress have paid little attention to the context of the reduced NADPH/NADP$^+$ ratio. Herein, we develop *Ec*STH to manipulate the homeostasis of NADH/NAD$^+$ and NADPH/NADP$^+$, and to mimic the NADH accumulation in reductive stress associated with a decreased NADPH/NADP$^+$ ratio. We have shown that the accumulated NADH in reductive stress can indirectly activate PRPS2 by antagonizing the inhibitory effect of its end product, and thus promote downstream purine biosynthesis, which could induce energy stress (Supplementary Fig. 17). Under normal physiological conditions, NADH acts as the fuel for mitochondrial ATP generation, and its content is an indicator of the state of the cell in term of energy levels. Therefore, it is reasonable to assume that cellular NADH triggers energy-consuming purine biosynthesis. Increased purine biosynthesis may activate mTORC1[23,28], and mTORC1, in turn, promotes ATP-driven anabolic pathways including purine nucleotide biosynthesis[29,30], constituting a positive regulation circle to provoke massive energy consumption. ADP, as one product derived

from the nucleotide biosynthesis pathway, can function as a potently physiological suppressor of PRPS1 and PRPS2, forming a negative feedback loop to regulate purine biosynthesis. In the presence of adequate ATP, NADH relieves PRPS2 from the inhibition afforded by ADP, thus leading to an increase in supplying PRPP for purine biosynthesis, which integrates ATP generation, nucleotide biosynthesis and cell proliferation. However, under reductive stress, the accumulated NADH is uncoupled from energy status and deregulates purine biosynthesis to initiate and deteriorate energy stress, which could contribute to reductive stress-induced cell death (Supplementary Fig. 17). Our current study has already demonstrated that the blockade of purine biosynthesis virtually attenuates energy crisis and cell death upon reductive stress.

On the other hand, we cannot exclude the possibility that reductive stress modulates cell death or tissue injury by regulating other pathways than purine biosynthesis. Meanwhile, we are also unable to rule out the possibility that reductive stress could simultaneously directly or indirectly regulate the activities of the downstream enzymes of PRPS1/2, which are involved in de novo purine biosynthesis. Nevertheless, the sensing pathway of NADH accumulation,

identified in our current study, appears to play a determinative role in the regulation of cell death associated with reductive stress in vitro and in vivo. Therefore, the control of the purine biosynthesis pathway could be a promising strategy to treat reductive stress-associated syndromes, such as ischemic injury, neurodegeneration, diabetes and mitochondrial diseases. For such purposes, 6-MP, a clinic drug, normally used to treat acute lymphocytic leukemia, Crohn's disease and ulcerative colitis[31–33], may represent an attractive candidate, because it is one of the safest and most effective medicines on the World Health Organization's list of essential medicines[34].

In addition, the engineered enzyme $Ec$STH-$Lb$NOX can substantially decrease the cellular NADPH and NADPH/NADP$^+$ ratio while leaving the cellular NADH and NADH/NAD$^+$ ratio relatively unaffected, rendering it a feasible research tool for the regulation of the cellular NADPH/NADP$^+$ ratio. For such a purpose, TPNOX, as well as its mitochondria-targeted form, mitoTPNOX, has been recently developed. They can oxidize NADPH to NADP$^+$ using oxygen as the elector acceptor[14]. However, TPNOX does not effectively change the ratio of NADPH/NADP$^+$, while mitoTPNOX can simultaneously reduce the ratios of both NADH/NAD$^+$ and NADPH/NADP$^+$. Therefore, we have provided additional genetic tools for use to investigate the homeostasis of NADH/NAD$^+$ or NADPH/NADP$^+$, which could be used as complementary tools with preexisting strategies. In the current study, we use $Ec$STH and $Ec$STH-$Lb$NOX to tease apart the roles of NADH/NAD$^+$ and NADPH/NADP$^+$, and demonstrate that cell growth, glycolysis and nucleotide biosynthesis are highly correlated with the homeostasis of NADH/NAD$^+$, whereas TCA metabolites seem to be instead associated with the homeostasis of NADPH/NADP$^+$. One may expect that $Ec$STH, in combination with $Ec$STH-$Lb$NOX, will be applied to animals, and bring us more and deeper in vivo insights related to NADH/NAD$^+$ and NADPH/NADP$^+$ in the future.

## Methods
### Cell lines and reagents
HeLa (CCL-2), MDA-MB-231 (HTB-26), and HEK293T (CRL-3216) cell lines were obtained from ATCC and were maintained in DMEM (Gibco, C11995500BT) with 10% fetal bovine serum (Gibco, 10099141 C) and 1% penicillin-streptomycin (Gibco, 15140-122) at 37 °C within a humidified atmosphere of 20% $O_2$ and 5% $CO_2$. Primary embryonic fibroblasts (MEFs) were prepared from E13.5 wild type C57BL/6 embryos using standard protocols[35] and maintained in DMEM with 10% fetal bovine serum. MEFs, cultured for no more than six passages, were used for experiments. Since pyruvate could attenuate the accumulation of NADH, we replaced the medium with pyruvate-free DMEM (Merck, D0819) in NADH-related experiments to avoid the interference of pyruvate. To induce hypoxia, cells were cultured under 0.5% oxygen conditions. All the cell lines were tested negative for mycoplasma, and no commonly misidentified cell lines were used in the study. Doxycycline (D9891), N-acetyl-L-cysteine (A9165), glutathione monoethyl ester (353905), and 2-ketobutyric acid (K401) were purchased from Sigma. 6-Mercaptopurine (HY-13677), pelitrexol (HY-14530), lometrexol (HY-14521B), Z-VAD-FMK (HY-16658B), necrosulfonamide (HY-100573), ferrostatin-1 (HY-100579), and menadione (HY-B0332) were purchased from MedChemExpress. A769662 (S2697) and AICAR (S1802) were purchased from Selleck.

### Plasmid constructs
Lentiviral Tet-On 3 G Inducible Expression Systems (Clontech) was used for the expression of $Ec$STH-Flag, Flag-$Lb$NOX and $Ec$STH-Flag-$Lb$NOX proteins. Humanized $Ec$STH-Flag cDNA were commercially synthesized by Genewiz. Flag-$Lb$NOX cDNA was a kind gift from Vamsi Mootha (Addgene, 75285). $Ec$STH-Flag-$Lb$NOX cDNA was created by PCR and verified by sequencing. The cDNAs were cloned into pLVX-TRE3G-vector using restriction digestion (BamHI and NotI). The cDNA for human TKT, PRPS1 and PRPS2 were amplified by PCR.

For overexpression, the cDNAs were cloned into pCDH vector, and for protein purification, the cDNAs were cloned into pET-28a vector. The LentiGuide-puro-Vector (Addgene, 52963) was used to generate sgRNA-expressing constructs. The sequences of sgRNA used in this study are listed as follows: sgTKT#1: CACCGATGGAAACGCCGCAG; sgTKT#2: CTCACAGATAGCCACCCGCA. PRPS1/2 small interfering RNAs (siPRPS1/2) were purchased from GenePharma. The target sequences of siRNAs used in this study are listed as the following: siPRPS1#1: CUGCACUAUUGUCUCACCUTT; siPRPS1#2: GUGAUUGACAUCUCUAUGATT; siPRPS1#3: CUUACCUAUUCAGCCAUGUTT; siPRPS2#1: GGAGACCAGCGUGGAGAUUTT; siPRPS2#2: CCGAGUGGAAGAACUGUAUTT; siPRPS2#3: GCUGUUGUCGUCACAAACATT. The target sequences of shRNA used in this study are listed as the following: shPRPS2: GCTGTAGGTATTCAGCAATGA.

### Lentivirus production and purification
For lentivirus package, two million HEK293T cells were seeded in a 100 mm dish with DMEM and 10% FBS. 24 h after seeding, 10 μg pCMV-dR8.91, 10 μg pCMV-VSVG, and 20 μg target plasmid were co-transfected into 293 T cells using 1 mg/mL PEI. The medium containing plasmid mixture and PEI was exchanged for fresh medium after incubation for 6 h. 48 h after transfection, the medium was collected and then concentrated using the Lenti-XTM concentrator kit (Clontech, 631231). The virus was resuspended using a DMEM medium and stored in aliquots at −80 °C.

### Immunoblotting
Cell pellets were lysed using lysis buffer containing 50 mM Tris-HCl (pH 7.4), 150 mM NaCl, 1 mM EDTA, 1% Nonidet P-40, 1 μg/mL aprotinin, 1 μg/mL leupeptin, and 1 μg/mL pepstatin, 1 mM Na3VO4 and 1 mM phenylmethylsulfonyl fluoride. The protein concentration was determined using the bicinchoninic acid assay (Beyotime Biotechnology, P0011) with a TriStar² LB 942 multimode microplate reader (Berthold Technologies). Samples were loaded into SDS-PAGE gels and then transferred to polyvinylidene fluoride membranes (Millipore). Membranes were blocked for 1 h in 5% milk in TBST solution at room temperature and then incubated in indicated primary antibodies at 4 °C overnight. After washing, the membranes were incubated with secondary antibodies for 1 h at room temperature before washing again and visualization with ECL (Millipore, WBKLS0500) (Supplementary Fig. 18). Anti-Flag (F1804, 1:5,000) antibody was purchased from Sigma. PRPS1 (15549-1-AP, 1:2,000), PRPS2 (27024-1-AP, 1:2,000), TKT (11039-1-AP, 1:5,000), GFP (50430-2-AP, 1:5,000), ACC1 (21923-1-AP, 1:2,000), GAPDH (60004-1-Ig, 1:5,000), PARP-1 (13371-1-AP, 1:1,000), Caspase-3 (19677-1-AP, 1:1,000), DHODH (14877-1-AP, 1:1000), Lamin B (12987-1-AP, 1:1000), αTubulin (11224-1-AP, 1:5,000), and βActin (66009-1-Ig, 1:5,000) antibodies were purchased from Proteintech. pACC1 (Ser79, 3661, 1:1,000), pAMPK (Thr172, 2535, 1:1,000), and AMPK (5831, 1:1,000) were obtained from CST. The goat anti-rabbit IgG/HRP (ZDR-5306, 1:5,000) and goat anti-mouse IgG/HRP (ZDR-5307, 1:5,000) secondary antibodies were obtained from ZSGB-Bio.

### Cell proliferation and apoptosis assay
Cell proliferation was measured by cell number. Briefly, cells were plated into a 24-well plate at 2000–5000 cells per well. The next day, a medium supplemented with doxycycline or water was added per well. Cells were counted every day for the time indicated in the experiments using Muse Cell Analyzer (Luminex Cooperation). Cell apoptosis rate was detected by Muse Annexin V & Dead Cell Kit (MCH100105, Luminex). Briefly, cells were plated into a 12-well plate at 50,000 cells per well. After treatment, all cells (including the upper floating dead cells) were collected and stained with 7AAD/Annexin V for 10 min at room temperature. Cell apoptosis rate was measured by Muse Cell Analyzer and analyzed by FlowJo 7.6.

## Clonogenic assays

Cells were plated on a 12-well plate at a density of 500 cells per well. The next day, a medium supplemented with doxycycline or water was added per well. After 10 days, the cells were washed twice with PBS, fixed, and stained with crystal violet solution for 10 min. The cells were washed with water, dried, and counted. Clone numbers were counted by Image J. The survival fraction was normalized to the number of water-treated groups.

## Measurement of NADH/NAD$^+$ and NADPH/NADP$^+$ ratios

Cellular NADH/NAD$^+$ and NADPH/NADP$^+$ ratios were measured using NADH-glo assay and NADPH-glo assay respectively (Promega, G9072 and G9082) according to the manufacturer's instructions. Briefly, $5 \times 10^4$ cells were seeded in a 24-well plate. 24 h after seeding, the medium was exchanged for DMEM with doxycycline or water. At the indicated time points after doxycycline addition, the cells were quickly washed with PBS and then 0.2 mL of an ice-cold 1:1 mixture of PBS and bicarbonate base buffer with 1% DTAB was added to lyse the cells. 50 μL of the cell lysate was transferred to an empty well of 96 well plates for the base treatment. For the acid treatment, the same volume of cell lysate and 25 μL of 0.4 N HCl were added. Both samples were incubated for 15 min at 60 °C to destroy reduced or oxidized nucleotides. 25 μL of 0.5 M TrizmaR base was added to each well of acid-treated cells to neutralize the acid and 50 μL of HCl/TrizmaR solution was added to each well containing base-treated samples. The cellular NADH/NAD$^+$ and NADPH/NADP$^+$ ratios were determined with the kits following the manufacturer's instructions, and the amounts of NADH, NAD$^+$, NADPH, or NADP$^+$ were normalized by cell numbers.

## Metabolite uptake and excretion

β-Hydroxybutyric acid (β-HB) and acetoacetic acid (AcAc) in the medium were detected by Ketone Body Assay Kit (Merck, MAK134), and pyruvate in the medium was using Pyruvate (PA) Assay Kit (Solarbio, BC2205) according to the manufacturer's instructions.

## GSH/GSSG ratio detection assay

Cellular GSH and GSSG were measured by GSH/GSSG Ratio Detection Assay (Abcam, ab205811). Briefly, $1 \times 10^4$ cells were seeded in a 24-well plate. 24 h after seeding, the medium was exchanged to DMEM with doxycycline or water to induce the expression of EGFP, *Lb*NOX, *Ec*STH, or *Ec*STH-Flag-*Lb*NOX for 24 h. Harvest $1 \times 10^4$ cells and wash cells with cold PBS twice. Resuspend cells in 50 μL PBS/0.5% NP-40 buffer and homogenize cells quickly by pipetting up and down a few times. Centrifugation at 14000 g for 15 min at 4 °C and transfer supernatant to a clean tube. Proteins from the sample were removed by TCA/NaHCO3 deproteinization and GSH/GSSG was measured following the manufacturer's instructions.

## *Lb*NOX, *Ec*STH and *Ec*STH-*Lb*NOX enzyme kinetics

*Lb*NOX, *Ec*STH, and *Ec*STH-*Lb*NOX were purified with ANTI-FLAG ® M2 Affinity Gel (A2220, Millipore) in HeLa cells expressing inducible *Lb*NOX-Flag, *Ec*STH-Flag, and *Ec*STH-Flag-*Lb*NOX after induction for 24 h (0.1 μg/mL Dox). Enzyme assays were performed in 100 μL assay buffer (50 mM Tris-HCl, pH 7.5, 10 μM flavin adenine dinucleotide, and NADH, NADPH, Thio-NAD + as indicated) at 35 °C for 5 min. Reduction of thio-NAD$^+$ was monitored at 405 nm and depletion of NADH (or NADPH) was assessed at 340 nm using a TriStar$^2$ LB 942 multimode microplate reader (Berthold Technologies).

## Transketolase activity measurement

The transketolase (TKT) activity was determined by Transketolase Activity Assay Kit (BioVision, K2004) according to the manufacturer's instructions. Briefly, $3 \times 10^5$ cells were planted into a 6-well plate. After 12 h of seeding, the medium was exchanged to DMEM with doxycycline for 24 h as indicated. Then rinsed with PBS twice and homogenized

cells with 80 μl TKT Assay buffer on ice for 10 min followed by centrifugation at 10,000 g and 4 °C for 15 min. The lysate supernatant was collected and the protein concentration was estimated using the bicinchoninic acid assay (Beyotime Biotechnology, P0011). 5 μg protein per group was used for measurement immediately. The fluorescence at Ex/Em 535/587 nm was measured using a TriStar$^2$ LB 942 multimode microplate reader (Berthold Technologies).

## Measurement of ATP levels

The intracellular ATP level was measured using a CellTiter-Glo Luminescent assay kit (Promega, G7570) according to the manufacturer's instructions. Briefly, Cells were seeded in a 96-well plate 1 day before indicated treatment. After the cells were incubated with the drugs for 2 h, doxycycline was added and cultured for the time indicated in the experiments. Then 100 μL CellTiter-Glo Reagent was added to each well, and then the contents were mixed and incubated at room temperature for 10 min on a shaker. The luminescence was measured using a TriStar$^2$ LB 942 multimode microplate reader (Berthold Technologies).

## Genome-wide CRISPR screen

The genome-wide CRISPR screen was performed in *Ec*STH cells with stably expressed Cas9. $1.67 \times 10^8$ cells were transduced with the human GeCKO v2 library (Addgene, 1000000049) or the Brunello library (Addgene, 73178) virus at a target MOI of 0.3. Briefly, 10 μl of concentrated GeCKO v2 library or the Brunello library was added to each plate containing $2 \times 10^6$ cells. 24 h after transduction, puromycin was added to the cells and maintained for 5 days. Cells were passaged as needed when reaching confluency. On day 5, cells were split into drug conditions in duplicate with $3 \times 10^7$ per replicate. Additionally, $5 \times 10^7$ cells were pelleted and frozen at −80 °C for genomic DNA extraction. Two replicates were cultured in DMEM with 100 ng/mL doxycycline whereas the other two replicates were cultured in DMEM with water. For the screen using the GeCKO v2 library, $5 \times 10^7$ cells were spun down and frozen for genomic DNA extraction after 3 days with water treatment or 7 days with doxycycline treatment. Cells were not passaged and the medium was exchanged every 3 days. For the screen using the Brunello library, cell pellets were taken at 9 days after water or doxycycline addition. Cells were passaged and the medium was exchanged every 3 days. The genomic DNA was extracted using the Blood & Cell Culture Midi kit (Qiagen, 13343). The sgRNA library readout was performed in two steps of PCR. For the first PCR, enough genomic DNA was used to ensure the full amplification of all the sgRNA harvested in the screen. Primers used for the first PCR were:

F1: 5′-AATGGACTATCATATGCTTACCGTAACTTGAAAGTATTTCG-3′,

R1: 5′-CTTTAGTTTGTATGTCTGTTGCTATTATGTCTACTATTCTTTCC-3′.

The PCR was performed using Herculase II Fusion DNA Polymerase (Agilent, 600679) and the resulting amplicons were pooled and used for amplification with barcoded second PCR primers. The second PCR products were gel extracted, quantified, mixed, and sequenced using a HiSeq-PE$^2$50. Significantly enriched or depleted sgRNAs were identified using MAGeCK software.

## Metabolite profiling and isotope tracing

LC/MS analysis was performed on a TSQ Quantiva triple quadrupole mass spectrometer coupled to a Dionex Ulti-Mate 3000 UPLC system (Thermo Fisher Scientific). Experiments were performed in a medium containing 10% dialyzed FBS (Gibco). DMEM lacking glucose, glutamine, and pyruvate was prepared from powder (Sigma), and then supplemented with labeled glucose or labeled glutamine. All the reconstituted experimental media finally contained 10 mM glucose and 1 mM glutamine. For *Ec*STH and *Ec*STH-*Lb*NOX cells, $2 \times 10^6$ cells were grown in 100-mm dishes, 12 h after seeding, the medium was

exchanged to DMEM with doxycycline or water for another 12 h to induce protein expression, then rinsed with PBS and cultured with 10 mL isotopes-containing medium for the time as indicated in the experiments. Cells were washed three times with PBS, and polar metabolites were extracted in 1.5 mL of 80% methanol (prechilled to -80 °C). After extraction, supernatants were nitrogen-dried and stored at -80 °C. Dried polar samples were resuspended in 50 µL 80% methanol, and 1 µL was injected into a Synergi Hydro-RP 100 A 2.1 × 100 mm column (Phenomenex) for metabolite separation with column temperature at 35 °C. Mobile phases A and B were 10 mM tributylamine in aqueous with pH 5 and 100% methanol, respectively. The chromatographic gradient was set for mobile phase B as follows: 0–3.5 min: 1% B; 3.5–22 min: from 1% to 70% B; 22–23 min: from 70% to 90% B; 23–25 min: 90% B; 25–30 min: 1% B. Data were acquired using the positive/negative switching method. Spray voltages of 3.5 and 2.5 kV were applied for positive and negative modes, respectively. Q1 and Q3 resolution were set at 0.7 and 1 s of cycle time was used in the method. Quantitation of polar metabolites was performed with Tracefinder (Thermo Fisher Scientific). Retention times and ion transitions of all metabolites were validated according to chemical standards. Ion pairs with various isotope labels were obtained based on precursors' and fragments' chemical structures. The abundance of each mass isotopomer was then mathematically corrected to eliminate natural abundance isotopes and finally converted into a percentage of the total pool.

## Targeted metabolomics

Samples were prepared as described in the section "Metabolite profiling and isotope tracing". For dried polar metabolites, the analysis was performed using TSQ Quantiva (Thermo, CA). C18-based reverse phase chromatography was applied. Mobile phase A is 10 mM tributylamine, and 15 mM acetate in HPLC-grade water. Mobile phase B is 100% HPLC-grade methanol. This analysis was focused on purine and pyrimidine metabolism, glycolysis, TCA cycle, pentose phosphate pathway, and amino acid metabolism. For this analysis, a 25-minute gradient from 5% to 90% mobile B was used. Data acquisition was obtained using the positive-negative ion switching mode. Spray voltages of 3.5 and 2.5 kV were applied for positive and negative modes, respectively. The resolution for Q1 and Q3 are both 0.7FWHM. The source mass spectrometer parameters are shown as follows: capillary temperature: 320 °C; heater temperature: 300 °C; sheath gas flow rate: 35; auxiliary gas flow rate: 10. Tracefinder (Thermo Fisher Scientific) search with home-built database containing about 300 compounds was used for metabolite identification. For fatty acid analysis, the samples were resuspended in 100 µl dichloromethane: methanol (v: v = 1:1) and prepared for LC/MS analyses. The UPLC system was coupled to a Q-Exactive HF orbitrap mass spectrometer (Thermo Fisher, CA). Cortecs C18 column (2.1 × 100 mm; Waters) was applied for analysis. Mobile phase A was ACN: H2O (60:40), 10 mM ammonium acetate. Mobile phase B was IPA: ACN (90:10). The gradient was as below: 0 min, 30% B; 2.5 min, 30% B; 8 min 50% B; 10 min, 98% B; 15 min 98% B; 15.1 min, 30% B; 18 min 30% B. The flow rate was 220 µL/min. Data with mass ranges of $m/z$ 150−600 was acquired at negative ion mode. The full scan was collected with a resolution of 60,000. The source parameters are as follows: spray voltage: 3.0 kV; capillary temperature: 320 °C; heater temperature: 300 °C; sheath gas flow rate: 35; auxiliary gas flow rate: 10. Tracefinder (Thermo Fisher, CA) search with endogenous MS database by accurate masses was used for lipid identification.

## Untargeted metabolomics

Samples were prepared as described in the section "Metabolite profiling and isotope tracing". Untargeted metabolomic analysis was performed using Q Exactive orbitrap (Thermo, CA). 1 µL supernatant was injected into a BEH Amide 100×2.1 mm column (Waters) for the

positive mode. Then, the samples were eluted to orbitrap mass spectrometer with aqueous containing 5 mM ammonium acetate as eluent from 1% to 99% within 15 min. The stationary phase was 95% acetonitrile with 5 mM ammonium acetate. In negative mode, the BEH C18 column (2.1 × 100 mm, Waters) was applied for analysis at a flow of 0.25 mL/min. Mobile phase A was made by mixing 1 L of HPLC-grade water containing 5 mM ammonium bicarbonate. Mobile phase B contained 100% ACN. Data with mass ranges of $m/z$ 70–1050 and $m/z$ 80–1200 was acquired at positive ion mode and negative ion mode with data-dependent MSMS acquisition. The full scan and fragment spectra were collected with the resolution of 70,000 and 17,500 respectively. The source parameters are as follows: spray voltage: 3.0 kV; capillary temperature: 320 °C; heater temperature: 300 °C; sheath gas flow rate: 35; auxiliary gas flow rate: 10. Metabolite identification was based on Tracefinder (Thermo Fisher Scientific) search with the home-built database. The database containing MS/MS standard spectra from over 1500 metabolites was applied for metabolite identification. The mass tolerances of 10 ppm and 15 ppm were used for precursor and fragment matching.

## PRPS1/2 purification and activity measurement

His-PRPS1 and His-PRPS2 were expressed in E.coli BL21 cells and purified using a $Ni^{2+}$−Sepharose affinity column (cytiva, 17-5248-01). NAD/NADH/GDP/ADP were diluted to the corresponding concentration with assay buffer (50 mM Tris, pH 7.5, 5 mM MgCl2, 5 mM Na3PO4, 1 mM DTT, 1 mM EDTA, 600 µM ATP, 600 µM R-5-P). Enzyme (50 mM Tris, pH 7.5, 5 mM MgCl2, 5 mM Na3PO4, 1 mM DTT, 1 mM EDTA, 2 µg/mL PRPS1 or 2) was blended with the mixture in equal volume for a final reaction system (50 mM Tris, pH 7.5, 5 mM MgCl2, 5 mM Na3PO4, 1 mM DTT, 1 mM EDTA, 300 µM ATP, 300 µM R-5-P, 1 µg/mL PRPS1 or 2). The reaction was incubated at 37°C for 10 min and terminated by adding an equal volume of 10 mM CaCl2. For the kinetics study, the activity of PRPS1/2 was measured based on ATP consumption that was detected according to the manufacturer's instructions of the Kinase-Glo Kit (Promega, V6072).

## Surface plasmon resonance analysis

SPR analysis was conducted with a Biacore T200 biomolecular interaction analysis system (GE Healthcare) following the manufacturer's protocol. Briefly, His-tagged PRPS2 was purified as described above and immobilized on the surface of a CM5 sensor chip in PBS buffer. The NADH or $NAD^+$ solution was diluted with running buffer PBS to the concentrations indicated in the experiments. The surface was regenerated using regeneration buffer (10 mM glycine, pH 2.0). The affinity parameters were calculated by Biacore T200 evaluation software.

## Molecular docking

3D structure of PRPS1 was downloaded from the RCSB PDB database (3S5J). The monomer of PRPS1 was extracted as the docking receptor. The 3D structure of PRPS2 was downloaded from AlphaFold (AF-P60891-F1). The 2D structure for the molecule ligands (ADP and NADH) was downloaded from the PubChem database. ChemBio 3D software was used to calculate and derive the 3D structure by minimizing energy. PyMOL 2.4.0 software was used to perform the dehydration of the receptor protein. Autodock software was used to carry out hydrogenation and charge calculation of proteins. Parameters of the receptor protein docking site were set to the whole protein. Finally, Autodock Vina was used to dock the receptor protein with the small molecule ligands. The best docking model with the lowest binding energy was selected.

## Acute ethanol gavage model

The animal protocol was approved by the IACUC at Capital Medical University in accordance with the principles and procedures outlined in the National Institutes of Health Guide for the Care and Use of

Laboratory Animals. Drugs were dissolved in a Vehicle buffer (10% DMSO, 40% PEG400, 50% saline). 6-MP (100 mg/kg) and pelitrexol (50 mg/kg) were pre-injected into the tail vein for 1 h for inhibition of purine biosynthesis. Lentivirus ($3 \times 10^8$ PFU) was pre-injected into the tail vein for 3 days for $Lb$NOX expression. Mice were sacrificed after ethanol (6 g/kg) gavage for 9 h. The activity of the serum ALT and AST were measured by blood sampling from the eyes. Cellular NADH/NAD$^+$ level, ATP, polar metabolites, and TUNEL assay were performed on the same liver lobe.

## Mice xenografts

All of the xenograft experiments were approved by the IACUC at Capital Medical University in accordance with the principles and procedures outlined in the National Institutes of Health Guide for the Care and Use of Laboratory Animals. Female nude mice (4–5 weeks) were purchased from Beijing Vital River Laboratory Animal Technology Co. Ltd. $3 \times 10^6$ HeLa$^{Tet-on \; EcSTH}$ or HeLa$^{Tet-on \; EcSTH-LbNOX}$ cells were injected subcutaneously into mice. When the tumor volume reached 100–200 mm$^3$ (10 days), the mice were assigned randomly into two groups ($n = 5$). The Dox treatment group received Dox (2 mg/mL) in their drinking water. The control group received distilled water throughout the study. The tumor volume was measured every 2 days until the endpoint and calculated using the formula $1/2 \times L \times W^2$.

## Statistical analysis

The results of all the experiments were generated from at least three independent replicates unless noted otherwise in the figure legend. All results are presented as mean ± standard deviation (SD). Plots and statistical analysis were performed using Prism 7 software (GraphPad). Statistical significance was calculated as appropriate using a two-tailed Student's $t$-test or two-way ANOVA as indicated in the figure legends. Metabolomics data were analyzed by MetaboAnalyst 5.0.

## Reporting summary

Further information on research design is available in the Nature Portfolio Reporting Summary linked to this article.

## Data availability

All data are available within the Article, Supplementary Information or Source Data file. Source data are provided with this paper.

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

## Acknowledgements

We thank Dr. Xiaohui Liu (Metabolomics Facility at Tsinghua University Branch of China National Center for Protein Sciences, China) for technical help, and Dr. Bin-Bing Zhou (Shanghai Jiao Tong University School of Medicine, China) for providing plasmids. This work was supported by Grants 81972567 and 82030093 from the Natural Science Foundation of China, Grant Jingyiyan-2021-10 from Beijing Municipal Institute of Public Medical Research Development and Reform Pilot Project, and Support Project for High-Level Teachers in Beijing Municipal Universities during the Period of 13th Five-Year Plan grant CIT&TCD20190333.

## Author contributions

B.L. conceived and designed the study and supervised the project; R.Y. and C.Y. performed experiments; L.M., Z.G., and Y.Z. created some constructs and cell lines; J.N., Q.C., and Y.M. provided advice; B.L., C.Y, and R.Y. analyzed the data; B.L. and C.Y. wrote the manuscript.

## Competing interests

The authors declare no competing interests.
