## [Peer Review File · Nature Communications]

Identification of purine biosynthesis as an NADH-sensing pathway to mediate energy stressReviewers' Comments:

Reviewer #1:

Remarks to the Author:

Comments for the Author

--How do you account for what appears to be stunning efficiency in the ability of the fusion protein (EcSTH-LbNOX) to restore the NADH/NAD⁺ ratio precisely to normal (no overshooting, e.g.)?

--Absolute (and total) adenine nucleotide levels should be provided for each set of conditions (i.e., NADH, NAD⁺, NADPH, and NADP⁺)

--How did GSH/GSSG ratios (as well as absolute and total GSH, GSSG levels) change under each set of conditions?

--Similar studies should be performed with non-immortalized, primary cells to demonstrate the generalizability of these mechanisms beyond immortalized/cancer cells.

--What effect did the TKT knock-out have on GSH/GSSG, and what were its consequences for cell death? How did 6MP, pelitrexol, and lometrexol affect this ratio? How did changes in this ratio affect the cell's susceptibility to additionally imposed redox stress (e.g., menadione) and its cytotoxic consequences?

--Assuming that GSH/GSSG was decreased concomitant with the NADPH/NADP⁺ decrease, could administration of glutathione monoethylester protect cells from injury/death?

--In the sgTKT volcano plot experiment shown in Figure 3e, how do you explain the increase in 6-phosphogluconate without a concomitant increase in glucose-6-phosphate (or its other downstream glycolytic metabolites)?

Reviewer #2:

Remarks to the Author:

The manuscript by Yang et. al. introduces a new tool to purportedly mimic reductive stress and studies its associated metabolic consequences. The authors express a soluble transhydrogenase from *E. coli* (EcSTH) with which to manipulate NADH/NAD⁺ and NADPH/NADP⁺ in the cytosol. Using this tool, the authors find that catalyzing hydride transfer from cytosolic NADPH to cytosolic NADH modulates PRPS2 activity, thereby upregulating purine biosynthesis. This maladaptive response then leads to an energy crisis and subsequent cell death. The authors recapitulate this finding in a mouse model of reductive stress. Further, they use LbNOX, a previously developed tool that specifically oxidizes NADH to NAD⁺, to determine that upregulation of purine biosynthesis is indeed due to increased NADH. The study reports an interesting application of soluble transhydrogenases in mammalian systems and I find the linkage to purine biosynthesis to be interesting and novel. There are aspects of the tool that are not presented appropriately.

There are several key questions that need to be addressed before the manuscript can be considered for publication.

Major comments:

1. The title and abstract claim that the authors have developed a clean genetic tool for inducing NADH reductive stress is simply not correct. Their tool transfers hydrides between NADH/NAD⁺ and NADPH/NADP⁺ and the directionality will be dependent on the prevailing ratios in the relevant compartment.

So their tool when expressed in the cytosol induces both NADH reductive stress *AND* NADPH oxidative stress. LbNOX and TPNOX -- on the other hand -- are cleaner tools with which to specifically alleviate NADH reductive stress or NADPH reductive stress. I think the authors need to shy away from the claim that their tool induces reductive stress -- it is not true -- rather - simply call it a soluble transhydrogenase and explain it causes NADH reductive stress and NADPH oxidative stress. (The combined use with LbNOX as a fusion is what allows them to tease apart the NADH versus NADPH role.)

2. In Fig. 1g, it is unclear why cytosolic EcSTH-induced reductive stress increases secreted beta-HB/AcAc, a marker of mitochondrial reductive stress, in HeLa which does not express BDH1/2. Can the authors explain this? This does not make any sense.

3. The media used for the cell culture growth assays and genome-wide CRISPR screen (Figs. 2b, c, f; 3b-d, g-h;) should be explicitly mentioned. Was pyruvate present in the medium? The methods indicate that DMEM with high glucose, pyruvate was used. If true, Figs. 2 b-c, 3b-d, 5b, S1, S4d-e don't make sense -- as pyruvate in the media should completely cover the NADH/NAD⁺ reductive stress in the cytosol.

4. The authors should test and discuss the relative kinetics of the EcSTH and LbNOX enzymes, and compare to the fused EcSTH-LbNOX enzyme.

5. The manuscript begins by characterizing reductive stress under Antimycin A treatment and hypoxic conditions. While these are complex perturbations, it is essential that the authors recapitulate the finding that purine biosynthesis is upregulated in these conditions as well.

6. A dysregulated increase in purine biosynthesis in response to EcSTH-induced reductive stress is shown to starve cells of ATP. This is further shown to decrease cell proliferation. While the authors repeatedly claim that this energy crisis causes cell death, this cell death is not characterized. The authors should perform an Annexin V/PI staining to characterize the type of cell death.

7. Data in second panel of S1a seems to have identical data under control conditions- is this in error?

Minor comments:

1. The referencing of the first 4-5 papers do not make sense. The manuscript begins with a discussion on reductive stress and tools with which to manipulate them -- appropriate papers should be cited here should include LbNOX as a tool and in vivo, as well as LOXCAT which also alleviates reductive stress. These come later -- but the initial 4-5 references don't make sense as presented.

2. Studies that biochemically characterized EcSTH (eg. Cao et al., 2011 and others) should be cited appropriately.

3. % oxygen for hypoxia should be mentioned in figure legend.

4. STH is misspelled as SHT in some figures (Eg. Fig. 1b, Fig. S2a-b). This should be corrected.

Reviewer #3:

Remarks to the Author:

Reductive stress is characterized by NADH accumulation and increase in the NADH/NAD⁺ ratio. There

is growing evidence that reductive stress is a hallmark of various pathologic disorders, however the tools selectively study the biological effects of NADH accumulation are limited. Here, Yang et al. describe a tool that enables the study of the cellular outcome of high NADH with low NADPH, and the combination of this tool with LbNOX to allow for separation between the NADH and the NADPH effects. The tool is based on the bacterial STH that pairs NAD conversion to NADH with NADPH conversion to NADP. Yang et al. find that STH expression causes cell death via NADH accumulation, which can be rescued with the exogenous electron acceptor alpha-ketobutyrate, validating ETC inhibition as the cause of the reduced cell fitness.

The authors used their tool in a genome-wide CRISPR screen and found that KO of the gene TKT, that is part of the PPP pathway that feeds into glycolysis (and not purine synthesis) increase STH-induced cell death.

Knockout of TKT is synergistic with reductive stress in causing cell death, highlighting increased purine biosynthesis as a potential mechanism for reductive stress-induced cell death.

Yang et al. performed metabolite profiling on TKT ko cells that indicated increased intermediates of the PPP and purine synthesis pathway. The data led the authors to postulate the purine synthesis pathway as a pathway that is responsive to reductive stress. Further, the authors show that negative regulation of the purine synthesis enzyme PRPS2 by ADP is alleviated by increased NADH concentrations. They therefore suggest that NADH accumulation causes deregulation of purine biosynthesis, which reduces total ATP levels and leads to energy crisis and cell death. This can be rescued using purine synthesis inhibitors and AMPK activators. Lastly, the authors induce reductive stress in livers by acute ethanol-induced hepatotoxicity and show that purine biosynthesis inhibitors reduce liver damage in this model.

The authors report a novel tool to experimentally manipulate and study the downstream effects of NADH accumulation. This tool can be used for the study of reductive stress in cells and potentially in tissues and sharing this tool with the cancer metabolism community can be very impactful. Their work highlights the novel role of NADH sensing in regulating the purine biosynthesis enzyme PRPS2 and identifies purine biosynthesis as a possible target to reduce reductive stress-induced cell death.

Major comments:

1. The paper describes a potential "sensing" of reductive stress by the purine synthesis pathway through PRPS2. However, most of the data presented in the paper supports the already-described inhibition of the TCA cycle by high NADH levels [PMID 22106302, 32187526, 22101431 and others]. For example, supplementary figure 6 strongly supports TCA cycle inhibition in the +DOX cells. The exciting discovery here can be that this is not mediated only through depletion of available NAD (as previously proposed), but also directly by high NADH. However, the authors should address TCA cycle flux in their experiments. The data for this should be readily available to them from their targeted metabolomics and C13-Glucose labeling experiments. Further, the inhibition of the TCA cycle as a plausible reason for increased dependency on glycolysis (as found in the screen) should be discussed thoroughly in the paper.

2. The authors suggest that depletion of ATP pools following induction of reductive stress is due to increased consumption by deregulated purine biosynthesis (page 7, line 21 of Results). However, mitochondrial and ETC dysfunction may also contribute to ATP/ADP ratio imbalance and ATP depletion. Mitochondrial dysfunction could be expected because of excess NADH equivalents being shuttled from cytosol to the mitochondria (or other downstream mechanisms of reductive stress). The authors could address the potential role of mitochondrial dysfunction by performing a Seahorse assay on EcSTH vs. EcSTH-LbNOX in expressing cells with and without doxycycline. Additionally, the alternative explanation of ATP depletion by low ETC function should be discussed in the interpretation of AMP activation (that is a direct sensor of ATP through AMP levels).

3. The metabolic outcome of NADH accumulation and the role of TKT should be better assessed by applying metabolite profiling with more tight controls: The metabolic consequences of high NADH should be tested in a comparison of EcSTH vs. EcSTH-LbNOX expressing cells. The role of TKT in reductively-stressed cells should be assessed by metabolomics of sgTKT vs. sgControl cells without EcSTH (no doxycycline) instead of the presented metabolomics comparison in Fig. 3e that is on sgTKT

and sgControl cells in the context of EcSTH expression.

4. The PRPS-targeting siRNAs used in Figures 4g-h and 5c should be rescued with PRPS expressing vectors to confirm on-target effect, especially since the efficiency of the knockdown is weak (Fig 4F).
5. Given the authors highlight the role of PRPS2 as a potential NADH sensor, some more information should be added to that section; how did this gene score in their CRISPR screens in Figure 3? Discuss the differences between PRPS1 and PRPS2 (that have 95.3% homology), and whether these differences can explain the differential response to NADH.
6. Figure 6 does not address the applicability of the EcSTH tool for work in vivo, although the authors have the ability to use exogenous expression systems, such as the one they used to express flag-LbNOX. The ethanol-induced reductive stress should be studied side-by-side with the EcSTH system expressed in hepatocytes (with and without DOX in the drinking water). Given the complexity of this experiment, I don't support requesting this for the revision, but I encourage an honest discussion of the data presented in figure 6 in light of this shortcoming.
7. Given the lethal consequences of inhibition of purine synthesis, the authors should include cell counts later than day 2 in figures 3 and 5 (for 6-MP, PTrexol and LTrexol), and figure 4 (for PRPS1/PRPS2/PRPS1+2 knockdown), and longer than 9 hours in figure 6. This will also allow consistency with figure 2e, where reductive stress caused cell death in a scale of 10-20 days.
8. A good optional addition to the previous comment can be metabolite profiling of cells treated with these harsh treatments (6-MP, PTrexol and LTrexol, PRPS1/PRPS2/PRPS1+2 knockdown) to show purine and purine-synthesis intermediates levels.
9. All labeling experiments (figures 4d, Supp 6) should also include the total levels of each of the presented metabolites.

Minor comments:

1. The title can be reworded so it is more descriptive of the presented findings or of the tool that is demonstrated here.
2. The manuscript would benefit from further review by an English editor, with particular attention to: the first and last paragraphs of the Introduction, Methods (Acute ethanol gavage model, mice xenografts sections), and various grammatical errors throughout the Results section.
3. References got disordered (citations in the text begin with #5).
4. The biological question could be more clearly stated in the Introduction.
5. In the introduction – a limitation of the LbNOX enzyme is mentioned (dependency on oxygen), but given the importance of LbNOX for the application of new tool presented here, the language should be modified. The strength and novelty of EcSTH is mostly in the metabolites it changes, and only mildly in its function independent of oxygen, because it requires the oxygen-consuming LbNOX for full interpretation of the results.
6. Figure 1b: add (+dox) to the "Time" label
7. Figure 2a: add vertical space between the blots that are from different cell lines
8. Please confirm that DMSO was the drug vehicle used in vivo drug treatments.
9. Lines 12-13 in page 5: the conclusion should be limited to cell lines: "Taken together, EcTSH can function as a genetic tool in cell lines to stimulate NADH-reductive stress.
10. Please add a scheme of TKT function in figure 3.
11. Figure 3c and f: add p-values for significance
12. In some figures the comparisons used for the various p values are not very clear and should be clarified in the legend (see for example Fig 3G,H).
13. Figure 4a: typo "Phosphocreatine1" ◊ "Phosphocreatine"
14. Figure 4a and supp 5f – the metabolite profiling heatmaps: Please check the color bar. Also – are the values presented a Z score or metabolite abundance? If Z-score – please also add metabolite abundance. If metabolite abundance – please correct the label on the color bar.
15. Figure 4d and e: for consistency, please change "STH-NOX" to "EcSTH-LbNOX"
16. The authors demonstrate that only PRPS2 is regulated by NADH (Fig. 4i, Supp 8b-c). It seems surprising that the proliferative phenotype is rescued by siPRPS1 alone if cells would become more dependent on the NADH responsive PRPS2, although this could be explained by decreased flux through purine biosynthesis. The authors could include relative expression data (by qPCR or western

blot) of PRPS1 and PRPS2 in ECSTH-expressing HeLa cells with and without DOX.

17. Figure 5e – please add metabolite profiling data of AICAR to exclude reduced AMPK activation due to reduced AICAR synthesis.

18. Figure 6f – the IF is not clear. The DAPI is very faint and therefore the TUNEL is hard to interpret.

19. Supp. Fig. 6: the scaling for the y axes for all graphs in this figure are unclear. For ease of interpretation, they could be normalized to the [m+0, -dox] condition.

20. Supp Fig. 7: please add NADH levels for LbNOX alone (in figure 1 NADH levels are shown only for EcSTH, and EcSTH-LbNOX).

21. Supp. Fig. 8b panel c has a typo “GPD \diamond GDP”

Dear all reviewers,

We greatly appreciate the constructive comments from the reviewers. Now we have revised the manuscript with further experiments and more accurate statements, which greatly improves the overall quality of our manuscript and makes the conclusion much more solid. The point-by-point responses to reviewers' comments can be found below (blue). All the changes in the manuscript are highlighted by blue.

Reviewer #1 (Remarks to the Author):

Comments for the Author

1,--How do you account for what appears to be stunning efficiency in the ability of the fusion protein (*Ec*STH-*Lb*NOX) to restore the NADH/NAD⁺ ratio precisely to normal (no overshooting, e.g.)?

Response: Thank you so much for pointing out this issue.

To address this issue, we carried out the experiments to investigate the kinetics of *Lb*NOX, *Ec*STH, and *Ec*STH-*Lb*NOX, and put these data as Figure 2a-g (new data in the current version). Our results showed that the V_{\max} and k_{cat} of *Lb*NOX is much greater than those of *Ec*STH (Fig. 2b-f in the current version, new data), so that *Ec*STH-*Lb*NOX was unable to produce NADH while only consumed NADPH (Fig. 2g in the current version, new data), consistent with the results of *Ec*STH-*Lb*NOX expressed in cells (Fig. 1f, and Supplementary Fig. 2f in the current version). Therefore, the NADH/NAD⁺ ratio is determined by *Lb*NOX when *Ec*STH-*Lb*NOX was used.

As for *Lb*NOX expression, it alone indeed did not affect the ratio of total NADH/NAD⁺ in cells and also left cell proliferation unaffected in the normal condition (*Science* 2016, 352(6282):231-5), which was also confirmed by our current results (Figure 3b,c in the current version). This was possibly attributed to the compartmental distribution or rapid compensation of NADH/NAD⁺ in the normal condition. Therefore, *Lb*NOX was widely used to attenuate NADH accumulation or reduce the increased ratio of NADH/NAD⁺ *in vitro* and *in vivo* (*Nature* 2020, 585(7824):288-292; *Nature* 2020, 583(7814):122-126).

Now we added the descriptions about these kinetics data in the part of "Generation of tools for manipulation of cellular NADH/NAD⁺ and NADPH/NADP⁺" in the results section: "We then purified *Ec*STH, *Lb*NOX, and *Ec*STH-*Lb*NOX proteins (Fig. 2a in the current version, new data), and measured their kinetic parameters. Our results showed that V_{\max} , K_M and k_{cat} values of *Ec*STH were comparable to those of *Ec*STH in *Ec*STH-*Lb*NOX (Fig. 2b-e in the current version, new data). However, V_{\max} and k_{cat} of *Lb*NOX were much greater than those of *Ec*STH (Fig. 2f in the current version, new data), so that *Ec*STH-*Lb*NOX was unable to produce NADH while only consumed NADPH (Fig. 2g in the current version, new data), consistent with the results obtained from *Ec*STH-*Lb*NOX expressed in cells (Fig. 1f and Supplementary

Fig. 2f)".

2,--Absolute (and total) adenine nucleotide levels should be provided for each set of conditions (i.e., NADH, NAD⁺, NADPH, and NADP⁺)

Response: Thank you so much for your suggestions.

Maybe you want to know the changes of NADH, NAD⁺, NADPH, or NADP⁺ under different conditions. In the last version, we measured the total contents of adenine nucleotides from pooled cells and normalized them by cell number. We plotted these data as relative NADH, NAD⁺, NADPH, or NADP⁺ to compare the relative changes in their total levels (associated with their ratio changes) under treatments. We put most of these data in the supplemental figures. However, in Fig. 1c,f, we used the relative ratios of NADH/NAD⁺ and NADPH/NADP⁺, and did not provide the relative levels of NADH, NAD⁺, NADPH, or NADP⁺ under the treatments. It might confuse and misguide you. Sorry for this. Now as you suggested, we replotted these data (Fig. 1c,f) with their real ratios and compared the relative changes in the total levels of NADH, NAD⁺, NADPH, and NADP⁺ (Supplementary Fig. 2b,f in the current version, new data). Meanwhile, we now provided all data about the changes of NADH, NAD⁺, NADPH, or NADP⁺ associated with their ratios as the supplementary data, including Supplementary Fig. 1a,b, 2b,c,d,f,g,h (new data), 3a,b, 4b,c, 5b,c (new data), and 15a,b (mouse data) in the current version.

3,--How did GSH/GSSG ratios (as well as absolute and total GSH, GSSG levels) change under each set of conditions?

Response: Thank you so much for your constructive suggestions.

As you suggest, we now measured GSH/GSSG ratios, as well as the absolute levels of total GSH and GSSG, and compared their changes. We added the results and related descriptions in the part of "*Ec*STH causes growth inhibition and cell death by inducing NADH accumulation" in the results section: "Intriguingly, our results showed that *Ec*STH-*Lb*NOX, *Ec*STH, or *Lb*NOX did not affect the ratios or contents of GSH and/or GSSG (Supplementary Fig. 6a in the current version, new data). An oxidative inducer, menadione, potently killed cells, but such an effect was largely reversed by the addition of GSH monoethylester or NAC (N-acetyl-L-cysteine, a precursor of GSH) (Supplementary Fig. 6b in the current version, new data). In sharp contrast, supplementation with GSH monoethylester or NAC did not rescue cell death induced by *Ec*STH at all (Supplementary Fig. 6b in the current version, new data). Meanwhile, we also found that cells with or without expression of *Ec*STH or *Ec*STH-*Lb*NOX had the similar sensitivity to menadione (Supplementary Fig. 6c in the current version, new data). These data suggested that oxidative stress was not involved in *Ec*STH-induced cell death. Although *Lb*NOX fusion almost completely restored the cellular level of NADH and the ratio of NADH/NAD⁺, it did not reverse the decreased ratio of NADPH/NADP⁺, with cellular NADPH being reduced to no less than 60% [Fig. 3b and Supplementary Fig. 3b, 4c and 5c (new data) in the current version]. However, *Ec*STH-*Lb*NOX did not influence cell growth *in vivo* and *in vitro* [Fig. 3c,e and Supplementary Fig. 4d and 5d (new data) in the current version]. Therefore, it appears that most likely, cellular NADPH or NADPH/NADP⁺, even declined to some extent, supports cell proliferation and survival sufficiently. These data suggest that NADH accumulation plays a critical role in *Ec*STH-induced cell growth arrest or cell death. Taken

together, *EcSTH* can function as a genetic tool to simulate NADH-accumulation alone or in combination with *LbNOX*".

Although it seems a little bit surprising, previous related reports did not directly prove the association between the homeostasis of cellular NADPH and GSH/GSSG. Most of such studies change cellular NADH/NADP⁺ or GSH/GSSG by altering metabolic transformations or oxidative stress. To my best knowledge, *EcSTH-LbNOX* is the only research tool to merely directly reduce NADPH and the ratio of NADPH/NADP⁺. A recently reported *TPNOX*, as well as its mitochondria-targeted form, *mitoTPNOX*, can oxidize NADPH to NADP⁺ using oxygen as the electron acceptor (*Nat Chem Biol* 13:1088-1095, 2017). However, *TPNOX* does not effectively change the ratio of NADPH/NADP⁺, while *mitoTPNOX* can simultaneously reduce the ratios of both NADH/NAD⁺ and NADPH/NADP⁺. Therefore, our current paper also developed *EcSTH-LbNOX* to study the role of NADPH, in addition to *EcSTH*. In fact, in our current study, *EcSTH-LbNOX* teased apart the changes of NADPH/NADP⁺ from the GSH/GSSG homeostasis.

4.--Similar studies should be performed with non-immortalized, primary cells to demonstrate the generalizability of these mechanisms beyond immortalized/cancer cells.

Response: Thank you so much for your constructive suggestions.

As you suggested, we now prepared primary embryonic fibroblasts (MEFs) from E13.5 wild type C57BL/6 embryos using standard protocols (Ref#36) and maintained MEFs in DMEM with 10% fetal bovine serum. MEFs, cultured for no more than six passages, were used for experiments (we described MEF preparation in the methods section). We expressed inducible *EcSTH* in MEF cells, and observed the similar results (Supplementary Fig. 5a-f and 11d in the current version, new data). MEF cells are unsuitable to be plated at low density, otherwise they will senesce faster. Thus, we did not conduct the time-course experiments of proliferation, which usually lasted for five days and required cell seeding at low density at the beginning. Instead, we measured the fold change increase in cell number in two days (Supplementary Fig. 5d,e and 11d). We added the results and related descriptions in the part of "*EcSTH* causes growth inhibition and cell death by inducing NADH accumulation" in the results section: "Now, to test the generality of *EcSTH* and *EcSTH-LbNOX*, we inducibly expressed them in another cancer cell line, MDA-MB-231, and mouse embryonic fibroblasts (MEF) [Supplementary Fig. 4a and 5a (new data) in the current version], and obtained the similar results. *EcSTH* expression increased the level of NADH and ratio of NADH/NAD⁺, while decreased the ratio of NADPH/NADP⁺ [Supplementary Fig. 4b,c and 5b,c (new data) in the current version]. *LbNOX* fusion restored the homeostasis of NADH/NAD⁺, but not that of NADPH/NADP⁺ (Supplementary Fig. 4b,c and 5b,c). Similarly, *EcSTH*, not *EcSTH-LbNOX*, suppressed cell proliferation or elicited cell death in MDA-MB-231 and MEF cells [Supplementary Fig. 4d and 5d (new data) in the current version]. The electron acceptor, α -ketobutyrate, also rescued proliferation (Supplementary Fig. 4e and 5e in the current version, new data) and prevented apoptosis of MDA-MB-231 and MEF cells induced by *EcSTH* expression (Fig. 3h,i and Supplementary Fig. 4f and 5f in the current version, new data)".

5.--What effect did the TKT knock-out have on GSH/GSSG, and what were its consequences for cell death? How did 6MP, pelitrexol, and lometrexol affect this ratio? How did changes in

this ratio affect the cell's susceptibility to additionally imposed redox stress (e.g., menadione) and its cytotoxic consequences?

Response: Thank you so much for your constructive suggestions.

As you suggested, we carried these experiments. Our results showed that TKT-KO did not affect GSH/GSSG ratios, as well as the absolute levels of total GSH and GSSG levels, in cells (see the attached data, Panel A). TKT-KO alone did not induce cell death (Fig. 4d). 6MP, pelitrexol and lometrexol also did not affect affect GSH/GSSG ratios, and the absolute levels of total GSH and GSSG levels (see the attached data, Panel B). Menadione killed cells, and under such conditions, which was largely reversed by the addition of GSH monoethyl ester or NAC (N-acetyl-L-cysteine, a precursor of GSH) (Supplementary Fig. 6b in the current version, new data). In sharp contrast, supplementation with GSH monoethyl ester or NAC did not rescue cell death induced by *EcSTH* at all (Supplementary Fig. 6b in the current version, new data). Meanwhile, we also found that the inhibitory capacity of menadione was similar for cells with or without expression of *EcSTH* or *EcSTH-LbNOX* (Supplementary Fig. 6c in the current version, new data). These data suggested that GSH/GSSG was not involved in *EcSTH*-induced cell death.

Legends: A, Effects of TKT knockout on cellular GSH and GSH/GSSG level in HeLa cells. B, Effects of purine biosynthesis inhibitors on cellular GSH and GSH/GSSG level in HeLa cells. Cells were treated with the indicated inhibitors for 24 h. 6-Mercaptopurine (6-MP, 300 μ M), pelitrexol (10 μ M), lometrexol (10 μ M).

6,--Assuming that GSH/GSSG was decreased concomitant with the NADPH/NADP⁺ decrease, could administration of glutathione monoethylester protect cells from injury/death?

Response: Thank you so much for your constructive suggestions.

As we responded in the above, GSH/GSSG was not decreased in *EcSTH*-expressing cells (Supplementary Fig. 6a in the current version, new data), although the cellular NADPH/NADP⁺ was significantly reduced (Fig. 3b and Supplementary Fig. 3b in the current version). Administration of glutathione monoethylester did not protect cells from injury/death (Supplementary Fig. 6b in the current version, new data). We explained the possible

mechanisms in the responses to Issue#3.

7,--In the sgTKT volcano plot experiment shown in Figure 3e, how do you explain the increase in 6-phosphogluconate without a concomitant increase in glucose-6-phosphate (or its other downstream glycolytic metabolites)?

Response: Thank you so much for your concerns.

In wild type cells (without TKT knockout), *EcSTH* expression increased metabolites in the PPP, glycolysis, and purine pathways (Fig. 5a, and Supplementary Fig. 9a in the current version). Now we carried out the experiments to investigate the effects of TKT-knockout alone on glycolytic metabolites and purines. Our results demonstrated that TKT-knockout alone also increased the cellular contents of purines, PPP and glycolytic metabolites (Supplementary Fig. 7c in the current version, new data). However, when cellular metabolites were profiled in HeLa^{Tet-on *EcSTH*}/sgControl and HeLa^{Tet-on *EcSTH*}/sgTKT cells in the presence of Dox, our results showed that TKT knockout further enhanced the cellular levels of purines but not those of glycolytic metabolites induced by *EcSTH* (Fig. 4f,g in the current version). Therefore, taken together, these data better support that TKT knockout-promoted cell death in HeLa^{Tet-on *EcSTH*} cells is highly associated with elevated purine biosynthesis.

We added these results and related descriptions in the part of “Elevated purine biosynthesis is associated with NADH accumulation-induced cell death” in the results section: “We now performed metabolic profiling on HeLa/sgControl and HeLa/sgTKT cells, and found that TKT-knockout increased the levels of cellular metabolites in the pathways of glycolysis, PPP, and purine biosynthesis (Supplementary Fig. 7c in the current version, new data). However, *EcSTH* expression did not affect the protein level or activity of TKT (Supplementary Fig. 7d in the current version), and exogenous over-expression of TKT did not prevent cell growth arrest or cell death induced by *EcSTH* expression (Supplementary Fig. 7e in the current version). Therefore, to determine how TKT blockade exacerbated cell death induced by *EcSTH* expression, we carried out a targeted metabolomics analysis on cells expressing *EcSTH* with or without TKT-knockout”.

Reviewer #2 (Remarks to the Author):

The manuscript by Yang et. al. introduces a new tool to purportedly mimic reductive stress and studies its associated metabolic consequences. The authors express a soluble transhydrogenase from *E. coli* (*EcSTH*) with which to manipulate NADH/NAD⁺ and NADPH/NADP⁺ in the cytosol. Using this tool, the authors find that catalyzing hydride transfer from cytosolic NADPH to cytosolic NADH modulates PRPS2 activity, thereby upregulating purine biosynthesis. This maladaptive response then leads to an energy crisis and subsequent cell death. The authors recapitulate this finding in a mouse model of reductive stress. Further, they use *LbNOX*, a previously developed tool that specifically oxidizes NADH to NAD⁺, to determine that upregulation of purine biosynthesis is indeed due to increased NADH. The study reports an interesting application of soluble transhydrogenases in mammalian systems and I find the linkage to purine biosynthesis to be interesting and novel. There are aspects of the tool that are not presented appropriately.

There are several key questions that need to be addressed before the manuscript can be considered for publication.

Major comments:

1. The title and abstract claim that the authors have developed a clean genetic tool for inducing NADH reductive stress is simply not correct. Their tool transfers hydrides between NADH/NAD⁺ and NADPH/NADP⁺ and the directionality will be dependent on the prevailing ratios in the relevant compartment.

So their tool when expressed in the cytosol induces both NADH reductive stress *AND* NADPH oxidative stress. *LbNOX* and *TPNOX* -- on the other hand -- are cleaner tools with which to specifically alleviate NADH reductive stress or NADPH reductive stress. I think the authors need to shy away from the claim that their tool induces reductive stress -- it is not true -- rather - simply call it a soluble transhydrogenase and explain it causes NADH reductive stress and NADPH oxidative stress. (The combined use with *LbNOX* as a fusion is what allows them to tease apart the NADH versus NADPH role.)

Response: Greatly grateful to you for the constructive suggestions.

I agreed with you. Now we changed the title to “Identification of purine biosynthesis as an NADH-sensing pathway to mediate energy stress”. As you pointed out, *EcSTH* increased the ratio of NADH/NAD⁺ while decreased that of NADPH/NADP⁺, and *LbNOX* fusion restored the increased ratio of NADH/NAD⁺ but not the decreased ratios of NADPH/NADP⁺. As for *EcSTH-LbNOX*, actually it is equal to that we use *LbNOX* to only restore the increased ratio of NADH/NAD⁺ while not to affect the decreased ratio of NADH/NAD⁺ induced by *EcSTH*. That is to say, in the context with a decreased NADPH/NADP⁺ ratio (corresponding to the stressful pathological conditions), we only restored the homeostasis of NADH/NAD⁺. By comparing *EcSTH* and *EcSTH-LbNOX*, we found that *EcSTH*-induced NADH accumulation accounted for cell death, and identified PRPS2-mediated purine biosynthesis as an NADH-sensing pathway to mediate energy stress. Based on the current topic, we modified the paper throughout to avoid claiming that our tool induced reductive stress.

2. In Fig. 1g, it is unclear why cytosolic EcSTH-induced reductive stress increases secreted beta-HB/AcAc, a marker of mitochondrial reductive stress, in HeLa which does not express BDH1/2. Can the authors explain this? This does not make any sense.

Response: Thank you so much for your concerns.

I am not sure whether BDH1/2 is a misspelling of "IDH1/2". To my best knowledge, HeLa cells have wild type IDH1/2, which was reported in many previous papers (PMID#29378948, 30903027 and 33536406), please see the attached data. In addition, ME1 and ME3 also can mediate the homeostasis of NADPH between the cytosol and mitochondrial matrix. Actually, beta-HB/AcAc is a marker of mitochondrial NADH accumulation.

Figure 1 A from PMID#29378948 (Proc Natl Acad Sci U S A, 2018, doi: 10.1073/pnas.1711257115). (A) Whole cell lysates from HeLa and HCT116 cells transduced with shctrl, shIDH1-AS1-1, or shIDH1-AS1-2 were subjected to Western blot analysis.

Figure 3i from PMID#30903027 (Nat Commun, 2019, doi: 10.1038/s41467-019-09352-1). i, Validation of the method based on knock-down of IDH1 or IDH2 genes and following citrate isotopic labeling after feeding HeLa cells with [U-13C]-glutamine.

Supplementary Figure 7a from PMID#33536406 (Signal Transduct Target Ther, 2021, doi: 10.1038/s41392-020-00399-x). a, Western blots to validate the double knockout of IDH1 and IDH2 in HeLa cells; NADH/NAD⁺ ratio in HeLa/Cas9 and HeLa^{DKO} cells treated without or with antimycin A (1 μM) for 4h; Survival of HeLa/Cas9 and HeLa^{DKO} cells treated without or with antimycin A (1 μM) for 24h, normalized to untreated cells.

3. The media used for the cell culture growth assays and genome-wide CRISPR screen (Figs. 2b, c, f; 3b-d, g-h;) should be explicitly mentioned. Was pyruvate present in the medium? The methods indicate that DMEM with high glucose, pyruvate was used. If true, Figs. 2 b-c, 3b-d, 5b, S1, S4d-e don't make sense -- as pyruvate in the media should completely cover the NADH/NAD⁺ reductive stress in the cytosol.

Response: Deeply appreciate you for your pointing out this issue.

I am sorry for this mistake. You are absolutely right. Keto acids including pyruvate and α -ketobutyrate can attenuate NADH accumulation, and in the current study, we already used α -ketobutyrate to reduce *Ec*STH-induced NADH accumulation (Supplementary Fig. 1a-c in the current version). Therefore, we used pyruvate-free medium to avoid the interference of pyruvate in all NADH-related experiments. Now we corrected the related descriptions in the methods section "Cell lines and reagents".

Thank you again!

4. The authors should test and discuss the relative kinetics of the *Ec*STH and *Lb*NOX enzymes, and compare to the fused *Ec*STH-*Lb*NOX enzyme.

Response: Thank you so much for your constructive suggestions.

As you suggested, now we measured the kinetics of *Ec*STH, *Lb*NOX and *Ec*STH in *Ec*STH-*Lb*NOX. Now we added the descriptions about these kinetics data in the part of "Generation of tools for manipulation of cellular NADH/NAD⁺ and NADPH/NADP⁺" in the results section: "We then purified *Ec*STH, *Lb*NOX and *Ec*STH-*Lb*NOX proteins (Fig. 2a in the current version, new data), and measured their kinetic parameters. Our results showed that V_{\max} , K_M and k_{cat} values of *Ec*STH were comparable to those of *Ec*STH in *Ec*STH-*Lb*NOX (Fig. 2b-e in the current version, new data). However, V_{\max} and k_{cat} of *Lb*NOX were much greater than those of *Ec*STH (Fig. 2f in the current version, new data), so that *Ec*STH-*Lb*NOX was unable to produce NADH while only consumed NADPH (Fig. 2g in the current version, new data), consistent with the results obtained from *Ec*STH-*Lb*NOX expressed in cells (Fig. 1f and Supplementary Fig. 2f in the current version)".

5. The manuscript begins by characterizing reductive stress under Antimycin A treatment and hypoxic conditions. While these are complex perturbations, it is essential that the authors recapitulate the finding that purine biosynthesis is upregulated in these conditions as well.

Response: Many thanks for your constructive suggestions.

Our lab works on cellular metabolism under hypoxia and electron transport chain (ETC) inhibition for many years, and frequently observed the increased levels of cellular purines. Here, we used ¹⁵N-amide-glutamine to label the *de novo* synthesized IMP, AMP and GMP. Our results demonstrated that hypoxia and ETC inhibition by antimycin A substantially increased the cellular levels of ¹⁵N-labelled IMP, AMP and GMP, which was significantly reversed by supplementation with α -ketobutyrate (Supplementary Fig. 10a-d in the current version, new data). We added the results and related descriptions in the part of "Purine biosynthesis functions as a NADH accumulation-sensing pathway" in the results section: "Furthermore, our results demonstrated that antimycin A and hypoxia increased the cellular levels of *de novo* synthesized purine nucleotides, as indicated by the labeling of IMP, AMP and GMP by ¹⁵N-amide-glutamine in HeLa and MDA-MB-231 cells (Supplementary Fig. 10a-d in the current version, new data).

Next, to determine whether NADH accumulation exerted a critical role in *de novo* purine biosynthesis under hypoxia and ETC inhibition, we supplemented cells with α -ketobutyrate to attenuate NADH accumulation. Our results showed that α -ketobutyrate significantly reduced the levels of ^{15}N -labeled and total IMP, AMP and GMP in HeLa and MDA-MB-231 cells under hypoxia or ETC inhibition (Supplementary Fig. 10a-d in the current version, new data), again supporting the contribution of NADH accumulation to purine biosynthesis”.

6. A dysregulated increase in purine biosynthesis in response to EcSTH-induced reductive stress is shown to starve cells of ATP. This is further shown to decrease cell proliferation. While the authors repeatedly claim that this energy crisis causes cell death, this cell death is not characterized. The authors should perform a Annexin V/PI staining to characterize the type of cell death.

Response: Thank you so much for the constructive suggestions.

As you suggested, now we carried out a series of experiments to demonstrate apoptosis induced by *EcSTH*. First, we used z-VAD-FMK (a pan-inhibitor of caspases-mediated apoptosis), necrosulfonamide (an inhibitor of necroptosis) and ferrostatin-1 (an inhibitor of ferroptosis) to treat cells expressing *EcSTH*. Our results showed that only z-VAD-FMK significantly prevented *EcSTH*-induced cell death (Fig. 3g in the current version, new data). Second, we performed Annexin V/7-AAD staining and measured the Annexin V-positive cells in *EcSTH* cells, which was protected by addition of α -ketobutyrate (Fig. 3h and Supplementary Fig. 3e,4f,5f in the current version, new data). Finally, we detected the cleaved caspase-3 and PARP1, further confirming the occurrence of apoptosis (Fig. 3i in the current version, new data).

We organized the results and related descriptions in the part of “*EcSTH* causes growth inhibition and cell death by inducing NADH accumulation” in the results section: “*EcSTH*-induced cell death was significantly prevented by z-VAD-FMK (a pan-inhibitor of caspases), but not by necrosulfonamide (an inhibitor of necroptosis) and ferrostatin-1 (an inhibitor of ferroptosis) (Fig. 3g in the current version, new data), suggesting the occurrence of apoptosis. It was further confirmed by the results that *EcSTH* expression obviously promoted Annexin V-positive cells (Fig. 3h and Supplementary Fig. 3e in the current version, new data), and enhanced the levels of cleaved caspase-3 and PARP1, which was prevented by addition of α -ketobutyrate (Fig. 3i in the current version, new data)”.

7. Data in second panel of S1a seems to have identical data under control conditions- is this in error?

Response: Thank you so much for your carefully reading our paper and pointing out the potential issues.

We checked the data, and found they are not identical and just happen to be very close. Please the data attached here.

	Relative NADH			Mean	SD
Control	0.9903623	1.067892	0.9417457	1	0.051948
α -KB	0.9921634	0.9379183	1.0699183	1	0.0541729

Minor comments:

1. The referencing of the first 4-5 papers do not make sense. The manuscript begins with a discussion on reductive stress and tools with which to manipulate them -- appropriate papers should be cited here should include LbNOX as a tool and in vivo, as well as LOXCAT which also alleviates reductive stress. These come later -- but the initial 4-5 references don't make sense as presented.

Response: Thank you so much for pointing out this error.

Now we already corrected it.

2. Studies that biochemically characterized EcSTH (eg. Cao et al., 2011 and others) should be cited appropriately.

Response: Thanks for your constructive suggestion.

As you suggested, now we now cited three references including Cao's paper (Ref16-18, PMID# 9098078, 9922271 and 21545646) when we first mentioned STH from *Escherichia coli* (EcSTH) or *Pseudomonas fluorescens* in the results section.

3. % oxygen for hypoxia should be mentioned in figure legend.

Response: Many thanks for your constructive suggestion.

Now we added the description about the oxygen content during hypoxia in the figure legends (Supplementary Fig. 1 and Fig. 10).

4. STH is misspelled as SHT in some figures (Eg. Fig. 1b, Fig. S2a-b). This should be corrected.

Response: Thank you so much for mentioning these issues.

Now we already corrected it.

Reviewer #3 (Remarks to the Author):

Reductive stress is characterized by NADH accumulation and increase in the NADH/NAD⁺ ratio. There is growing evidence that reductive stress is a hallmark of various pathologic disorders, however the tools selectively study the biological effects of NADH accumulation are limited. Here, Yang et al. describe a tool that enables the study of the cellular outcome of high NADH with low NADPH, and the combination of this tool with lbNOX to allow for separation between the NADH and the NADPH effects. The tool is based on the bacterial STH that pairs NAD conversion to NADH with NADPH conversion to NADP. Yang et al. find that STH expression causes cell death via NADH accumulation, which can be rescued with the exogenous electron acceptor alpha-ketobutyrate, validating ETC inhibition as the cause of the reduced cell fitness.

The authors used their tool in a genome-wide CRISPR screen and found that KO of the gene TKT, that is part of the PPP pathway that feeds into glycolysis (and not purine synthesis) increase STH-induced cell death.

Knockout of TKT is synergistic with reductive stress in causing cell death, highlighting increased purine biosynthesis as a potential mechanism for reductive stress-induced cell death. Yang et al. performed metabolite profiling on TKT ko cells that indicated increased intermediates of the PPP and purine synthesis pathway. The data led the authors to postulate the purine synthesis pathway as a pathway that is responsive to reductive stress. Further, the authors show that negative regulation of the purine synthesis enzyme PRPS2 by ADP is alleviated by increased NADH concentrations. They therefore suggest that NADH accumulation causes deregulation of purine biosynthesis, which reduces total ATP levels and leads to energy crisis and cell death. This can be rescued using purine synthesis inhibitors and AMPK activators. Lastly, the authors induce reductive stress in livers by acute ethanol-induced hepatotoxicity and show that purine biosynthesis inhibitors reduce liver damage in this model.

The authors report a novel tool to experimentally manipulate and study the downstream effects of NADH accumulation. This tool can be used for the study of reductive stress in cells and potentially in tissues and sharing this tool with the cancer metabolism community can be very impactful. Their work highlights the novel role of NADH sensing in regulating the the purine biosynthesis enzyme PRPS2 and identifies purine biosynthesis as a possible target to reduce reductive stress-induced cell death.

Major comments:

1. The paper describes a potential “sensing” of reductive stress by the purine synthesis pathway through PRPS2. However, most of the data presented in the paper supports the already-described inhibition of the TCA cycle by high NADH levels [PMID 22106302, 32187526, 22101431 and others]. For example, supplementary figure 6 strongly supports TCA cycle inhibition in the +DOX cells. The exciting discovery here can be that this is not mediated only through depletion of available NAD (as previously proposed), but also directly by high NADH.

However, the authors should address TCA cycle flux in their experiments. The data for this should be readily available to them from their targeted metabolomics and C13-Glucose labeling experiments. Further, the inhibition of the TCA cycle as a plausible reason for increased dependency on glycolysis (as found in the screen) should be discussed thoroughly in the paper. Response: Thank you so much for your constructive suggestions.

You are right. We had the data of TCA cycle metabolites. *EcSTH* indeed reduced the cellular levels of TCA cycle metabolites (see the attached data). However, *EcSTH-LbNOX* fusion restored the increased metabolites in the glycolytic pathway and purine biosynthesis, but did not reverse the decreased TCA cycle metabolites (see the attached data of *EcSTH-LbNOX*, compared with *EcSTH*). *LbNOX* alone did not affect the contents of these metabolites (see the attached data). Considering that *EcSTH-LbNOX* fusion restored the increased NADH/NAD⁺ ratio, but not the decreased NADPH/NADP⁺ ratio induced by *EcSTH*, it seemed that the decreased NADPH/NADP⁺ ratio is highly associated with the decreased TCA cycle metabolites in cells expressing *EcSTH* or *EcSTH-LbNOX*. Most likely, when intracellular NADPH was reduced, more glucose was completely oxidized to generate NADPH through the cycle of oxPPP and non-oxPPP, in order to maintain the intracellular NADPH at a relatively safe level. Therefore, when we profiled the differential metabolites between HeLa/*EcSTH* and HeLa/*EcSTH-LbNOX* cells, TCA cycle metabolites were not enriched (Fig. 5a).

Cells expressing *EcSTH-LbNOX* have relatively normal NADH/NAD⁺ homeostasis, glycolysis, purine biosynthesis, ATP levels, cell proliferation, tumor growth, and so on (Fig. 3, Fig. 5 and Fig. 6), but the NADPH/NADP⁺ ratio and TCA cycle metabolites were reduced. Thus, we did not know the exact biological functions of the reduced NADPH/NADP⁺ ratio and TCA cycle metabolites. As such, we used *EcSTH* and *EcSTH-LbNOX* to tease apart the roles of NADH/NAD⁺ and NADPH/NADP⁺, and provided some interestingly metabolic insights that we did not have the opportunity to observe before.

Legends: The cellular levels of TCA cycle metabolites determined by LC-MS in HeLa cells expressing inducible Tet-on *LbNOX*, *EcSTH* or *EcSTH-LbNOX* cultured with Dox (0.1 μg/ml) for 24 h.

Overall, *Ec*STH increased the ratio of NADH/NAD⁺ while decreased that of NADPH/NADP⁺, and *Lb*NOX fusion restored the increased ratio of NADH/NAD⁺ but not the decreased ratios of NADPH/NADP⁺. As for *Ec*STH-*Lb*NOX, actually it is equal to that we use *Lb*NOX to only restore the increased ratio of NADH/NAD⁺ while not to affect the decreased ratio of NADH/NAD⁺ induced by *Ec*STH. That is to say, in the context with a decreased NADPH/NADP⁺ ratio (corresponding to the stressful pathological conditions), we only restored the homeostasis of NADH/NAD⁺. By comparing *Ec*STH and *Ec*STH-*Lb*NOX, we found that *Ec*STH-induced NADH accumulation accounted for cell death, and identified PRPS2-mediated purine biosynthesis as an NADH-sensing pathway to mediate energy stress. Based on the current topic, we modified the paper throughout to avoid claiming that our tool induced reductive stress. Since the changes in TCA cycle metabolites induced by *Ec*STH and *Ec*STH-*Lb*NOX are similar, the data of TCA cycle metabolites do not help to improve our current manuscript. To avoid confusing readers, we did not put these data in the manuscript.

2. The authors suggest that depletion of ATP pools following induction of reductive stress is due to increased consumption by deregulated purine biosynthesis (page 7, line 21 of Results). However, mitochondrial and ETC dysfunction may also contribute to ATP/ADP ratio imbalance and ATP depletion. Mitochondrial dysfunction could be expected because of excess NADH equivalents being shuttled from cytosol to the mitochondria (or other downstream mechanisms of reductive stress). The authors could address the potential role of mitochondrial dysfunction by performing a Seahorse assay on *Ec*STH vs. *Ec*STH-*Lb*NOX in expressing cells with and without doxycycline. Additionally, the alternative explanation of ATP depletion by low ETC function should be discussed in the interpretation of AMP activation (that is a direct sensor of ATP through AMP levels).

Response: Thanks for the constructive suggestions.

As you suggested, we now performed the Seahorse assay on *Ec*STH cells. Since *Lb*NOX consumes oxygen, *Ec*STH-*Lb*NOX cannot be used to investigate OCR. We measured the OCR in HeLa/Tet-on EGFP and HeLa/Tet-on *Ec*STH in the presence or absence of Dox. These cells displayed similar OCR, suggesting that NADH accumulation possibly does not significantly impair the ETC (see the attached data). In fact, we found that *Ec*STH, hypoxia and ECT inhibition in cells, and *in vivo* ethanol administration in mice mainly increased the content of NADH but affected the cellular level of NAD⁺ to a lesser extent (Supplementary Fig. 1a, 3a, 4b, 5b and 15a in the current version). This may explain why mitochondrial OCR was not influenced. In addition, our results showed that knockdown of PRPS2 with siRNA or shRNA almost completely restored cellular ATP contents in cells expressing *Ec*STH (Fig. 6c in the current version, containing new data), also suggesting the promoted purine biosynthesis is mainly responsible for ATP depletion. However, in addition to the energy directly consumed by the process of purine biosynthesis, purines probably exerted other roles in provoking energy dissipation. For example, it was reported that increased purine biosynthesis can activate mTORC1, and mTORC1 in turn promotes ATP-driven anabolic pathways. Now we added some explanations in the discussion section: “Therefore, it is reasonable to assume that cellular NADH triggers energy-consuming purine biosynthesis. Increased purine biosynthesis may activate mTORC1 (Ref#22 and 27), and mTORC1 in turn promotes ATP-driven anabolic pathways including purine nucleotide biosynthesis (Ref#28 and 29), constituting a positive

regulation circle to provoke massive energy consumption. ADP, as one product derived from the nucleotide biosynthesis pathway, can function as a potently physiological suppressor of PRPS1 and PRPS2, forming a negative feedback loop to regulate purine biosynthesis”.

Legends: *Oxygen consumption rate (OCR) of HeLa cells expressing inducible EGFP or EcSTH was measured with Seahorse under treatment with 0.1 µg/ml of doxycycline. Data are the mean ± SD for triplicate experiments.*

3. The metabolic outcome of NADH accumulation and the role of TKT should be better assessed by applying metabolite profiling with more tight controls: The metabolic consequences of high NADH should be tested in a comparison of EcSTH vs. EcSTH-LbNOX expressing cells. The role of TKT in reductively-stressed cells should be assessed by metabolomics of sgTKT vs. sgControl cells without EcSTH (no doxycycline) instead of the presented metabolomics comparison in Fig. 3e that is on sgTKT and sgControl cells in the context of EcSTH expression.

Response: Greatly grateful to for the constructive suggestions.

As you suggested, now we performed the metabolomics of sgTKT vs. sgControl cells without *EcSTH*. Our results demonstrated that TKT-knockout alone also increased the cellular contents of metabolites in PPP, purine biosynthesis and glycolysis (Supplementary Fig. 7c in the current version, new data). When cellular metabolites were profiled in HeLa^{Tet-on EcSTH}/sgControl and HeLa^{Tet-on EcSTH}/sgTKT cells in the presence of Dox, our results showed that TKT knockout further enhanced the cellular levels of purines but not those of glycolytic metabolites induced by *EcSTH* (Fig. 4f,4g and Supplementary Fig. 7g in the current version). Taken together, these data better prove that TKT knockout-promoted cell death in HeLa^{Tet-on EcSTH} cells is highly associated with elevated purine biosynthesis.

In wild type cells (without TKT knockout), *EcSTH* expression increased metabolites in the PPP, glycolysis, and purine pathways (Fig. 5a in the current version). As you suggested, we now carried out the experiments to investigate the effects of TKT-knockout alone on glycolytic metabolites and purines in HeLa cells. Our results demonstrated that TKT-knockout alone also increased the cellular contents of purines, PPP and glycolytic metabolites (Supplementary Fig. 7c in the current version, new data). However, when cellular metabolites were profiled in HeLa^{Tet-on EcSTH}/sgControl and HeLa^{Tet-on EcSTH}/sgTKT cells in the presence of Dox, our results showed that TKT knockout further enhanced the cellular levels of purines but not those of glycolytic metabolites induced by *EcSTH* (Fig. 4f,g). Taken together, these data better support

that TKT knockout-promoted cell death in HeLa^{Tet-on EcSTH} cells is highly associated with elevated purine biosynthesis. Therefore, we kept these results in cells with TKT-KO and *EcSTH* expression (Figure 3e in the last version, Fig. 4f in the current version), and put new metabolomics data of sgTKT vs. sgControl HeLa cells without *EcSTH* as Supplementary Fig. 7c (in the current version, new data).

We added these results and related descriptions in the part of “Elevated purine biosynthesis is associated with NADH accumulation-induced cell death” in the results section: “We now performed metabolic profiling on HeLa/sgControl and HeLa/sgTKT cells, and found that TKT-knockout increased the levels of cellular metabolites in the pathways of glycolysis, PPP, and purine biosynthesis (Supplementary Fig. 7c in the current version, new data). However, *EcSTH* expression did not affect the protein level or activity of TKT (Supplementary Fig. 7d in the current version), and exogenous over-expression of TKT did not prevent cell growth arrest or cell death induced by *EcSTH* expression (Supplementary Fig. 7e in the current version). Therefore, to determine how TKT blockade exacerbated cell death induced by *EcSTH* expression, we carried out a targeted metabolomics analysis on cells expressing *EcSTH* with or without TKT-knockout”.

4. The PRPS-targeting siRNAs used in Figures 4g-h and 5c should be rescued with PRPS expressing vectors to confirm on-target effect, especially since the efficiency of the knockdown is weak (Fig 4F).

Response: Thank you so much for the constructive suggestions.

It is difficult to restore PRPS expression in transient silencing experiments with siRNA. As you suggested, we now used shRNA to target the 3'UTR of PRPS2 and constructed a stable cell line with knockdown of PRPS2 (Fig. 5l in the current version, new data). We obtained similar results in PRPS2 knockdown cells and found that the cell death and ATP crisis was induced again after PRPS2 was re-expressed in PRPS2-knockdown cells (Fig. 5m and 6c in the current version, new data).

We added the results and related descriptions in the part of “NADH indirectly activates PRPS2” in the results section: “We further stably knocked down PRPS2 in HeLa^{EcSTH} cells and also re-expressed PRPS2 to restore it in knockdown cells (Fig. 5l in the current version, new data). As expected, *EcSTH* expression induced NADH accumulation regardless of the states of PRPS2 (Supplementary Fig. 12e in the current version, new data), but it only drastically promoted the cellular levels of *de novo*-synthesized and total IMP, AMP, and GMP, and elicited death in the PRPS2-expressing cells while not in the knockdown cells (Fig. 5m,n, and Supplementary Fig. 12f in the current version, new data). Interestingly, under the normal condition, PRPS2 knockdown did not affect the biosynthesis of IMP, AMP, and GMP, but almost completely prevented *EcSTH*-promoted purine biosynthesis (Fig. 5n in the current version, new data). Therefore, the elevated NADH level can potentially activate PRPS2 by relieving it from the physiological feedback inhibition afforded by its end product.”.

5. Given the authors highlight the role of PRPS2 as a potential NADH sensor, some more information should be added to that section; how did this gene score in their CRISPR screens in Figure 3 (1)? Discuss the differences between PRPS1 and PRPS2 (that have 95.3% homology), and whether these differences can explain the differential response to NADH (2).

Response: Thank you so much for mentioning these issues.

(1) TKT and PRPS2 exerted the opposite roles in *Ec*STH-induced cell death. The negative screening in our current study is based on the rationale that knocking out some gene can promote cell death of HeLa cells expressing Tet-on *Ec*STH induced by 0.1 $\mu\text{g}/\text{ml}$ of Dox. Therefore, TKT was hit in the top ten in both methods, while PRPS2 scored 6974th (among 19113 genes in GeCKOV2 library) and 7707th (among 20441 genes in Brunello library), respectively in both screens.

*Ec*STH expression by 1 $\mu\text{g}/\text{ml}$ of Dox killed cells by accumulating NADH, so we first tried to use the positive screening to identify the NADH-biosensor/pathway associated with cell death. During positive screening, cells would survive if the genes encoding biosensors were knocked out. If works, the positive screening is much easier than the negative screening used in our current paper. After screening for weeks, we enriched *Ec*STH-expressing cells that completely survived under induction by 1 $\mu\text{g}/\text{ml}$ of Dox. However, we did not detect *Ec*STH expression and found that *Ec*STH genes was actually deleted in the enriched cells. Since the library contains more than 120,000 sgRNAs, *Ec*STH gene should be knocked out by off-target of some sgRNAs after long time cell culture. Therefore, the positive screening could be unsuitable for the genetically-encoded tools-based models.

(2) As you suggested, we now provided docking data to explain the difference between PRPS1 and PRPS2: “We next tried to conduct molecular docking analysis to identify the binding sites of NADH and ADP in the predicted structure of PRPS2 by AlphaFold from UniProt. The docking results showed that NADH-binding site and ADP-binding site in PRPS2 were indeed overlapped (Supplementary Fig.13 a-c in the current version, new data). In addition, we also docked NADH and ADP on PRPS1 using its crystal structure, and found that ADP-binding site of PRPS1 shared critical amino acid residues with that of PRPS2. Although a NADH-binding site was also predicted in PRPS1, but it was very different from ADP-binding site ((Supplementary Fig.13d-f in the current version, new data). However, all the amino acid residues contributing to the binding of ADP and NADH to PRPS1 or PRPS2 are among the identical residues of both PRPS1 and PRPS2. Meanwhile, based on the structures, we noticed that all the different amino acid residues sporadically distributed on the surface of proteins. Therefore, they most likely affected the binding of NADH to PRPS2 by subtly changing the structure” in the part of “NADH indirectly activates PRPS2” in the results section.

6. Figure 6 does not address the applicability of the *Ec*STH tool for work in vivo, although the authors have the ability to use exogenous expression systems, such as the one they used to express flag-LbNOX. The ethanol-induced reductive stress should be studied side-by-side with the *Ec*STH system expressed in hepatocytes (with and without DOX in the drinking water). Given the complexity of this experiment, I don't support requesting this for the revision, but I encourage an honest discussion of the data presented in figure 6 in light of this shortcoming.

Response: Deeply grateful to you for your suggestion and understanding.

As you suggested, we now added some discussions in the last paragraph in the discussion section: “Therefore, we have provided additional genetic tools for use to investigate the homeostasis of NADH/NAD⁺ or NADPH/NADP⁺, which could be used as complementary tools with preexisting strategies. In the current study, we demonstrate their application in the cultured cells. One may expect that *Ec*STH, in combination with *Ec*STH-LbNOX, will be

applied to animals, and bring us more and deeper *in vivo* insights related to NADH/NAD⁺ and NADPH/NADP⁺ in the future”.

7. Given the lethal consequences of inhibition of purine synthesis, the authors should include cell counts later than day 2 in figures 3 and 5 (for 6-MP, PTrexol and LTrexol), and figure 4 (for PRPS1/PRPS2/PRPS1+2 knockdown), and longer than 9 hours in figure 6. This will also allow consistency with figure 2e, where reductive stress caused cell death in a scale of 10-20 days.

Response: Thank you for the concerns.

Maybe the unclear labeling of Fig. 2e (now Fig. 3 in the current version) misguide this reviewer. In the first 10 days, we did not carry out any treatment to make tumors grow, and mice were administered with Dox (2 mg/ml) in the drinking water at day 10 (the control group receiving distilled water throughout the experiment). In fact, we observed significant volume reduction after Dox induction only for 2 days. Now we bolded the addition of “Dox” in Fig. 3e (in the current version).

Inhibition of *de novo* purine synthesis does not lead to cell death at least during 2-3 days, which may be due to the compensation of purine synthesis in the salvage pathway. In our cell culture system, *EcSTH* expression was able to apparently induce cell death only in 24 hours, so it was reasonable to measure the rescuing effect of purine biosynthesis inhibition on cell death at the time of 2 days.

In figure 6 (now Fig. 7 in the current version), alcohol was metabolized rapidly in the liver and produced a large amount of NADH. By contrast, it needs to take some time to induce *EcSTH* expression. Some literatures had reported that alcohol gavage caused liver injury within 9 h (PMID #23449255, #12937155, and #22213272). Therefore, we performed the experiments after 9 hours of alcohol gavage, and importantly, our results demonstrated that attenuating NADH accumulation by *LbNOX* and blocking purine biosynthesis reduced cellular purines and prevented liver injury.

8. A good optional addition to the previous comment can be metabolite profiling of cells treated with these harsh treatments (6-MP, PTrexol and LTrexol, PRPS1/PRPS2/PRPS1+2 knockdown) to show purine and purine-synthesis intermediates levels.

Response: Thanks for your constructive suggestions.

We performed the metabolite profiling of HeLa^{Tet-on EcSTH} cells treated with purine synthesis inhibitors or PRPS2-KD, and found that the synthesis of purine (IMP, AMP, GMP) were significantly inhibited by purine biosynthesis inhibitors or PRPS2 KD, in the presence or absence of *EcSTH* expression (Fig. 5n, and Supplementary Fig. 11a,b and 12f in the current version, new data). These results further helped us to understand that it is indeed the inhibition of purine synthesis that rescued the occurrence of cell death. Unfortunately, no purine synthetic metabolites before IMP was detected in such a LC/MS method targeting IMP, AMP and GMP. However, the ¹³C₅-glutamine-labeled IMP, AMP and GMP are sufficiently to support *de novo* biosynthesis of purines.

9. All labeling experiments (figures 4d, Supp 6) should also include the total levels of each of the presented metabolites.

Response: Thanks for your constructive suggestions.

As you suggested, we now added the total levels of each of the presented metabolites in all labeling experiments.

Minor comments:

1. The title can be reworded so it is more descriptive of the presented findings or of the tool that is demonstrated here.

Response: Thanks for the constructive suggestion.

I agreed with you. Now we changed the title to “Identification of purine biosynthesis as an NADH-sensing pathway to mediate energy stress”.

2. The manuscript would benefit from further review by an English editor, with particular attention to: the first and last paragraphs of the Introduction, Methods (Acute ethanol gavage model, mice xenografts sections), and various grammatical errors throughout the Results section.

Response: Thank you for the suggestion.

In fact, we already asked an editing company to modify our manuscript before submission, but it seemed that they did not change it too much. We now substantively modified the paper again, and also asked an English native speaker to polish it throughout, based on your suggestions.

3. References got disordered (citations in the text begin with #5).

Response: Thanks for pointing out this issue.

Now we have corrected it.

4. The biological question could be more clearly stated in the Introduction.

Response: Thanks for the constructive suggestions.

We now modified the biological question as “However, how accumulated NADH triggers pathogenic outcomes still remains to be elucidated. Hence, if we can determine effector pathways associated with reductive stress or NADH accumulation, we may provide more targets for treatment of these types of metabolic disorders” in the introduction section.

5. In the introduction – a limitation of the LbNOX enzyme is mentioned (dependency on oxygen), but given the importance of LbNOX for the application of new tool presented here, the language should be modified. The strength and novelty of EcSTH is mostly in the metabolites it changes, and only mildly in its function independent of oxygen, because it requires the oxygen-consuming LbNOX for full interpretation of the results.

Response: Thanks for the constructive suggestions.

You are right, we now deleted this sentence, and modified it as the biological question: “However, how accumulated NADH triggers pathogenic outcomes still remains to be elucidated”.

6. Figure 1b: add (+dox) to the “Time” label

Response: Thank you so much for mentioning this issue.

Now we already added “Dox (0.1 µg/ml)” to Fig. 1b and Supplementary Fig. 2a in the current version.

7. Figure 2a: add vertical space between the blots that are from different cell lines

Response: Thank you so much for mentioning this issue.

Now we already added vertical space between the blots in Fig. 3a (original Fig. 2a) and Supplementary Fig. 4a (in the current version).

8. Please confirm that DMSO was the drug vehicle used in vivo drug treatments.

Response: Thank you so much for mentioning this issue.

In fact, the inhibitors were pre-dissolved in DMSO at a high concentration. When administering drugs to animals, we prepared the drugs with 10% DMSO (for control) or 10% inhibitors (pre-dissolved in DMSO), 40% PEG400, 50% saline. Now we stated them in Methods and changed the label as “Vehicle” (now Fig. 7 in the current version).

9. Lines 12-13 in page 5: the conclusion should be limited to cell lines: “Taken together, EcTSH can function as a genetic tool in cell lines to stimulate NADH-reductive stress.

Response: Thank you so much for the suggestion.

Now we already corrected it to “Taken together, *EcSTH* can function as a genetic tool in cell lines to simulate NADH-accumulation alone or in combination with *LbNOX*”.

10. Please add a scheme of TKT function in figure 3.

Response: Thank you so much for the suggestion.

As you suggested, we now added a scheme of TKT function as Fig. 4c (Fig.3 was now changed to Fig. 4 in the current version).

11. Figure 3c and f: add p-values for significance

Response: Thank you so much for pointing out this issue.

Now we already added the p-values in these figures, now Fig. 4d and Supplementary Fig. 7b (in the current version).

12. In some figures the comparisons used for the various p values are not very clear and should be clarified in the legend (see for example Fig 3G,H).

Response: Thank you for pointing out the issues.

We re-annotated the p-values in the figures.

13. Figure 4a: typo “Phosphocreatine1” ◊ “Phosphocreatine”

Response: Thank you so much for your careful reading our manuscript and pointing out this error.

Now we already corrected it (Fig. 5a in the current version).

14. Figure 4a and supp 5f – the metabolite profiling heatmaps: Please check the color bar. Also – are the values presented a Z score or metabolite abundance? If Z-score – please also add metabolite abundance. If metabolite abundance – please correct the label on the color bar.

Response: Many thanks for your pointing out these issues.

I do not know why the color of the bar was incorrectly changed during document conversion (from Word to PDF). The values in Fig. 4a and Supplementary Fig. 5g (in the last version, now Fig. 5a and Supplementary Fig. 7g) were presented as a Z score. The abundance of main metabolites in Fig.4a (now Fig. 5a in the current version) was listed as Supplementary Fig. 9a,b (in the current version), and the metabolite abundance for Fig.S5f (now Supplementary Fig. 7g in the current version) was shown in Fig. 4f (in the current version). We now described the relationship between them in the legends.

15. Figure 4d and e: for consistency, please change “STH-NOX” to “EcSTH-LbNOX”

Response: Thank you so much for pointing out these issues.

Now we already corrected them in Fig. 4d,e (now Fig. 5c,e in the current version).

16. The authors demonstrate that only PRPS2 is regulated by NADH (Fig. 4i, Supp 8b-c). It seems surprising that the proliferative phenotype is rescued by siPRPS1 alone if cells would become more dependent on the NADH responsive PRPS2, although this could be explained by decreased flux through purine biosynthesis. The authors could include relative expression data (by qPCR or western blot) of PRPS1 and PRPS2 in EcSTH-expressing HeLa cells with and without DOX.

Response: Thank you so much the constructive suggestion.

As your suggestion, we measured the expression of PRPS1 and PRPS2 by western blot in *EcSTH* and *EcSTH-LbNOX* expressing HeLa cells (Supplementary Fig. 12a in the current version, new data). Our results showed that the protein levels of PRPS1 and PRPS2 have no change in *EcSTH* and *EcSTH-LbNOX* expressing HeLa cells. As you pointed out, knocking down PRPS1 also decreased the flux of purine biosynthesis, although NADH activated PRPS2.

17. Figure 5e – please add metabolite profiling data of AICAR to exclude reduced AMPK activation due to reduced AICAR synthesis.

Response: Thank you so much for the constructive suggestions.

We now used the commercial AICAR as the standard metabolite, and detected the abundance of cellular AICAR with LC/MS. As expected, cellular AICAR was significantly reduced and not detected after cells were treated with purine biosynthesis inhibitors (Supplementary Fig. 14b in the current version, new data). Therefore, the reduced AMPK activation might partly result from the reduced AICAR. However, we also observed that the levels of intracellular ATP were significantly restored after treatments with these inhibitors (Fig. 6d in the current version). So this does not affect our conclusion. We now added these results and descriptions in the part of “Purine biosynthesis contributes to energy stress-mediated cell death induced by NADH accumulation” in the results section: “Notably, the intermediate, AICAR, involved in the purine biosynthetic pathway could potentially activate AMPK, and it was also promoted by *EcSTH* and suppressed by the purine biosynthesis inhibitors (Supplementary Fig. 14b in the current version, new data). Therefore, to determine whether energy stress played a critical role in *EcSTH*-induced cell death, we pre-activated AMPK in cells with its activators, AICAR and A769662 (Supplementary Fig. 14c in the current version), to attenuate energy stress. We indeed observed that the two activators significantly prevented

ATP loss (Fig. 6f in the current version), and blocked cell death induced by *EcSTH* (Fig. 6g and Supplementary Fig. 14d in the current version). These results suggest that energy stress is the cause of *EcSTH*-induced cell death, while the concomitant AMPK activation possibly only functions as the negative feedback regulator”.

18. Figure 6f – the IF is not clear. The DAPI is very faint and therefore the TUNEL is hard to interpret.

Response: Thank you so much for mentioning this issue.

This may be caused by picture distortion during document conversion as the review materials. Now we adjusted the contrast of these images, please see the attached pictures here. It seems clear now (Fig. 7f in the current version, also see the attached data).

19. Supp. Fig. 6: the scaling for the y axes for all graphs in this figure are unclear. For ease of interpretation, they could be normalized to the [m+0, -dox] condition.

Response: Thank you so much for your constructive suggestions.

As your suggestion, the total levels of the metabolites were attached. However, the scaling for the Y axes in Fig.S6 (now Supplementary Fig. 8 in the current version) is normalized to the [Total, -dox] condition now, because in some cases, the level of metabolite m+0 in the absence of dox (m+0, -dox) is very low.

20. Supp Fig. 7: please add NADH levels for LbNOX alone (in figure 1 NADH levels are shown only for *EcSTH*, and *EcSTH*-LbNOX).

Response: Thank you so much.

The NADH levels for *LbNOX* alone are shown in Fig.3b and Supplementary Fig. 3a (for HeLa cells) and Supplementary Fig. 4b,c (for MDA-MB-231 cells) in the current version.

21. Supp. Fig. 8b panel c has a typo “GPD \diamond GDP

Response: Thank you so much for your careful reading our paper and pointing out this error.

Now we already corrected it (Supplementary Fig. 12d in the current version).

Reviewers' Comments:

Reviewer #1:

None

Reviewer #2:

Remarks to the Author:

The authors have addressed most of our concerns and the manuscript has been greatly improved by the addition of comparative kinetics of EcSTH and LbNOX and by re-focusing the title on the biology rather than the technology as a "clean" tool for manipulating reductive stress. The description of EcSTH along with EcSTH-LbNOX as a tool to simultaneously manipulate NADH/NAD⁺ and NADPH/NADP⁺ is now well-demonstrated and the downstream consequence of purine biosynthesis being affected is well-described.

However, there are still some major concerns that remain unaddressed -- these are largely concerns we raised in the initial round, and the authors have claimed to have addressed them, but they have not.

1. BDH1/2 catalyze the following reaction in the mitochondria, typically in hepatocytes:

Therefore, bHB/Acac is thought to be proportional to mitochondrial NADH/NAD⁺. bHB/Acac can only be a reliable measure of mito NADH stress when BDH1/2 are expressed, which is typically in hepatocyte mitochondria. Without BDH1/2 expression, this ratio does not make any sense and is not reflective of mito NADH/NAD⁺. The authors need to either show BDH expression or verify that the metabolites being detected are indeed bHB and Acac. (For clarity we are not confusing this enzyme with IDH.)

2. We made recommendations on sharpening the opening references 1-5, which ought to be focused on reductive stress, the authors state in their rebuttal they fixed this, but they have not.

3. In Fig. 2g, why is LbNOX unable to consume thioNADH? This does not make any sense since the enzyme seems active in Fig. 2f. The authors need to explain this.

4. In Fig. 3b, it seems that NADPH/NADP⁺ ratio is perturbed more with EcSTH-LbNOX than with EcSTH. Given that the two enzymes have comparable kinetic parameters, can the authors comment on whether this due to expression levels of the proteins or local/diffusion-limited NAD⁺ from LbNOX?

5. Given that the authors perform multiple metabolite measurements, a careful discussion on metabolites that change under EcSTH, ETC-inhibited or hypoxic conditions but not under EcSTH-LbNOX must be included, i.e., metabolites more susceptible to NADPH/NADP⁺ oxidative stress rather than NADH/NAD⁺ reductive stress, potentially proline, TCA metabolites, lipids etc.

6. In the abstract and in the introduction, LbNOX is described as a water-forming NADH oxidase. It would be appropriate to add that this tool has been previously described and characterized in detail E.g. in the abstract "... we fuse EcSTH with previously described LbNOX, a water-forming NADH oxidase ...".

Reviewer #3:

Remarks to the Author:

The authors have addressed our concerns for the most part.

The Seahorse results the authors presented in their rebuttal letter (major comment 2) are particularly surprising because changes in ATP and NADH levels, that the authors observed following DOX treatment, should result in different OCR, and yet the +/- DOX samples show identical OCR profile. We encourage the authors to look into this result again, and consider validating that the same cells exhibit the other described changes (eg ATP and NADH) following DOX treatment while showing no change in OCR.

Reviewer #1 states in Remark to Editor section that (s)he is satisfied with the revision.

Response: We are happy to learn that the Reviewer is satisfied with our revision. Greatly grateful to this reviewer again.

Reviewer #2 (Remarks to the Author):

The authors have addressed most of our concerns and the manuscript has been greatly improved by the addition of comparative kinetics of EcSTH and LbNOX and by re-focusing the title on the biology rather than the technology as a "clean" tool for manipulating reductive stress. The description of EcSTH along with EcSTH-LbNOX as a tool to simultaneously manipulate NADH/NAD⁺ and NADPH/NADP⁺ is now well-demonstrated and the downstream consequence of purine biosynthesis being affected is well-described.

However, there are still some major concerns that remain unaddressed -- these are largely concerns we raised in the initial round, and the authors have claimed to have addressed them, but they have not.

1. BDH1/2 catalyze the following reaction in the mitochondria, typically in hepatocytes:

Therefore, bHB/Acac is thought to be proportional to mitochondrial NADH/NAD⁺. bHB/Acac can only be a reliable measure of mito NADH stress when BDH1/2 are expressed, which is typically in hepatocyte mitochondria. Without BDH1/2 expression, this ratio does not make any sense and is not reflective of mito NADH/NAD⁺. The authors need to either show BDH expression or verify that the metabolites being detected are indeed bHB and Acac. (For clarity we are not confusing this enzyme with IDH.)

Response: We appreciate this reviewer for the concern and should say sorry for our misunderstanding the question raised by the reviewer in the last round of review.

As suggested, we now examined the protein expression of BDH1/2 in HeLa and AML-12 cells (mouse hepatocytes as the positive control) and found that BDH1 and BDH2 were expressed in both cell lines (see the attached data, Panel A). We also analyzed the expression level of BDH1 and BDH2 mRNA in more than 1000 cell lines in CCLE (Cancer Cell Line Encyclopedia) database and found that the mRNA of BDH1 and BDH2 were expressed in most cell lines including HeLa cells (see the attached data, Panel B). Since the expression of BDH1 and BDH2 in HeLa cells do not help to improve our current manuscript, we did not put these data in the manuscript to avoid confusing readers.

Legends: A, Western blots for the expression of BDH1 and BDH2 in AML-12 and HeLa cells. (BDH1, 15417-1-AP and BDH2, 27207-1-AP were purchased from Proteintech); B, The expression level of BDH1 and BDH2 mRNA in more than 1000 cell lines in CCLF database.

2. We made recommendations on sharpening the opening references 1-5, which ought to be focused on reductive stress, the authors state in their rebuttal they fixed this, but they have not. Response: Greatly grateful to you for pointing out this issue.

I am sorry for this. It seems that we deleted some contents, resulting in the incorrect citations, when we prepared this manuscript. Now we cited five references related to reductive stress, and deleted the original literatures. Again, I would like to appreciate you so much for this.

3. In Fig. 2g, why is LbNOX unable to consume thioNADH? This does not make any sense since the enzyme seems active in Fig. 2f. The authors need to explain this.

Response: Many thanks for pointing out our mistake.

We made an error in the Y-axis title of Fig. 2g. The Y-axis title on the left should be “Thio-NADH production” (not Thio-NADH consumption in the last version) and the right Y-axis title should be “NADPH consumption” (not NADPH production in the last version). The substrates involved in the enzymatic reaction of Fig. 2g were thio-NAD⁺ and NADPH. We evaluated the reaction by measuring the production of thio-NADH and the consumption of NADPH. LbNOX alone did not produce thio-NADH or consume NADPH. Instead, it effectively eliminated thio-NADH generated by EcSTH, so that we could not detect the thio-NADH produced by EcSTH in EcSTH-LbNOX. However, we still detected the consumption of NADPH by EcSTH in EcSTH-LbNOX. Now we have corrected this mistake. Thanks again for your help.

4. In Fig. 3b, it seems that NADPH/NADP⁺ ratio is perturbed more with EcSTH-LbNOX than with EcSTH. Given that the two enzymes have comparable kinetic parameters, can the authors comment on whether this due to expression levels of the proteins or local/diffusion-limited NAD⁺ from LbNOX?

Response: Thank you so much for your mentioning this issue.

Although the kinetic parameters of EcSTH was comparable to those of EcSTH in EcSTH-LbNOX, the NADH generated by EcSTH can be immediately consumed by the fused LbNOX, which may influence the equilibrium between the substrate, NADPH, and the product, NADP⁺,

of *Ec*STH. For sure, this could be also attributed to expression levels of the proteins or local/diffusion-limited NAD^+ from *Lb*NOX.

5. Given that the authors perform multiple metabolite measurements, a careful discussion on metabolites that change under *Ec*STH, ETC-inhibited or hypoxic conditions but not under *Ec*STH-*Lb*NOX must be included, i.e., metabolites more susceptible to NADPH/NADP⁺ oxidative stress rather than NADH/NAD⁺ reductive stress, potentially proline, TCA metabolites, lipids etc.

Response: Thanks a lot for the suggestions.

I agreed with you. Based on our current study, the cellular metabolites seemed to show differential response to the disturbance of NADH/NAD⁺ and NADPH/NADP⁺. Cells expressing *Ec*STH had the increased NADH/NAD⁺ ratio, reduced NADPH/NADP⁺ ratio, decreased cell proliferation, and enhanced glycolysis and nucleotide biosynthesis. By contrast, cells expressing *Ec*STH-*Lb*NOX had relatively normal NADH/NAD⁺ homeostasis, glycolysis, purine biosynthesis, ATP levels, cell proliferation, tumor growth, and so on, but the NADPH/NADP⁺ ratio and TCA cycle metabolites were reduced.

Now we added the data about TCA cycle metabolite as Supplementary Fig. 10, and described them as: “Notably, *Ec*STH reduced the cellular levels of tricarboxylic acid (TCA) cycle metabolites (Supplementary Fig. 10). However, *Ec*STH-*Lb*NOX fusion did not reverse the decreased TCA cycle metabolites (Supplementary Fig. 10). Therefore, the decreased TCA cycle metabolites seemed to be unrelated to cell growth in these cells” in the part “Purine biosynthesis functions as an NADH accumulation-sensing pathway” in the results section.

As you suggested, we now added some safe discussions: “Therefore, we have provided additional genetic tools for use to investigate the homeostasis of NADH/NAD⁺ or NADPH/NADP⁺, which could be used as complementary tools with preexisting strategies. In the current study, we use *Ec*STH and *Ec*STH-*Lb*NOX to tease apart the roles of NADH/NAD⁺ and NADPH/NADP⁺, and demonstrate that cell growth, glycolysis and nucleotide biosynthesis are highly correlated with the homeostasis of NADH/NAD⁺, whereas TCA metabolites seem to be instead associated with the homeostasis of NADPH/NADP⁺” in the last of the discussion section.

6. In the abstract and in the introduction, *Lb*NOX is described as a water-forming NADH oxidase. It would be appropriate to add that this tool has been previously described and characterized in detail E.g. in the abstract “.. we fuse *Ec*STH with previously described *Lb*NOX, a water-forming NADH oxidase ...”.

Response: Grateful to you for the constructive suggestion.

Now we added the description about *Lb*NOX in the abstract: “Furthermore, we fuse *Ec*STH with previously described *Lb*NOX (a water-forming NADH oxidase from *Lactobacillus brevis*) to resume the NADH/NAD⁺ ratio.”

Reviewer #3 (Remarks to the Author):

The authors have addressed our concerns for the most part.

The Seahorse results the authors presented in their rebuttal letter (major comment 2) are particularly surprising because changes in ATP and NADH levels, that the authors observed following DOX treatment, should result in different OCR, and yet the +/- DOX samples show identical OCR profile.

We encourage the authors to look into this result again, and consider validating that the same cells exhibit the other described changes (eg ATP and NADH) following DOX treatment while showing no change in OCR.

Response: Deeply grateful to you for the suggestions.

It is not possible to use the same cells for the detection of OCR, ATP and NADH, because these experiments are cell-destroying. Therefore, we already re-performed the Seahorse assay on *EcSTH* cells, and checked the cellular NADH/NAD⁺ and ATP in parallel under the same conditions. Our results showed that like EGFP expression (see the attached data, Panel A), *EcSTH* expression did not change the OCR (see the attached data, Panel B). Meanwhile, the cellular NADH/NAD⁺ was significantly elevated while the intracellular ATP content was significantly reduced after Dox treatment in HeLa^{Tet-on *EcSTH*} cells (see the attached data, Panel C and D). In fact, we measured the OCR of HeLa^{Tet-on *EcSTH*} cells under induction by 0, 0.01, 0.1 or 1 µg/ml of Dox, and did not observe the significant effects. We originally expected to observe an increase in OCR in the *EcSTH*-expressing cells, because they had more NADH contents. It appears that the mitochondrial respiration in HeLa cells is saturated, so that the mitochondrial OCR remains unchanged when intracellular NADH is further elevated.

Legends: **A,B**, Oxygen consumption rate (OCR) of HeLa cells expressing inducible EGFP (**A**) or *EcSTH* (**B**) was measured with Seahorse under treatment with 0.1 µg/ml of doxycycline. **C**, The ratios of cellular NADH/NAD⁺ in HeLa^{Tet-on *EcSTH*} cells cultured with Dox (0.1 µg/ml) for 12 h. **D**, Cellular ATP level of HeLa^{Tet-on *EcSTH*} cells cultured with Dox (0.1 µg/ml) for 12 h. Data are the mean ± SD for triplicate experiments.

Reviewers' Comments:

Reviewer #2:

Remarks to the Author:

The authors have now addressed my major critiques.

Reviewer #3:

Remarks to the Author:

The authors have addressed our latest concern and I find the paper suitable for publication in Nature Communications

Point-by-point responses (Round#3)

Reviewer #2 (Remarks to the Author): The authors have now addressed my major critiques.

Response: We thank the reviewer for his/her time reviewing the manuscript. Grateful to this reviewer again.

Reviewer #3 (Remarks to the Author): The authors have addressed our latest concern and I find the paper suitable for publication in Nature Communications

Response: We thank the reviewer for his/her time reviewing the manuscript. Grateful to this reviewer again.